# Tractable Expected Information Gains for Exponential Family Posteriors

**Rik Knowles** [1]    **Tom Rainforth** [1]

## Abstract

We investigate which models admit a collapse of the expected information gain (EIG) and its derivative from a doubly intractable to a singly intractable expression. We prove that a sufficient condition is that the posterior distribution belongs to an exponential family (EF) and depends on the experimental design and data only through its natural parameters, and derive corresponding singly intractable and unbiased estimators for the EIG and its (reparameterized) gradient. We further show that this is achieved when using a likelihood of an analogous form and any arbitrary prior. This is complemented by a theoretical analysis of certain degenerate behaviors that may arise when optimizing the EIG for EF-modeled experiments. Finally, we empirically demonstrate the benefits of using our singly intractable estimators, showing superior convergence rates, and substantial performance gains for sequential design problems compared to using standard nested estimators.

## 1. Introduction

Bayesian Experimental Design (BED) provides a powerful framework for optimizing experimental designs, by maximizing the expected information gain (EIG) about a parameter of interest (Lindley, 1956; 1972; Bernardo, 1979; Huan et al., 2024; Rainforth et al., 2024). However, despite its principled foundations, the practical deployment of BED is often hindered by its computational costs (Foster et al., 2021; 2019). In particular, the EIG of a design is a doubly-intractable quantity, taking the form of a nested expectation (Rainforth et al., 2018; 2024). Similarly, its gradient is doubly intractable unless exact samples can be directly drawn from the posterior (Ao & Li, 2023; Iollo et al., 2025).

It is well known that in special cases, most notably Gaussian likelihoods with Gaussian priors, the EIG admits closed-form or singly-nested expressions (Houlsby et al., 2011). However, a more general characterization of when the information gain becomes tractable has been lacking.

In this work, we provide such a characterization. We show that *whenever the posterior distribution belongs to an exponential family (EF) that depends on the design only through the natural parameters, the* EIG *collapses from a doubly-nested to a singly-nested expectation.* Under these conditions, we derive unbiased, singly-intractable estimators for both the EIG and its (reparameterized) gradient, which yield substantial computational savings over standard nested Monte Carlo methods.

Noting that this requirement concerns only the posterior itself, with no direct assumptions on the likelihood or prior, we then investigate when such a posterior form occurs. We see that the key requirement is that the likelihood lives in a fixed EF for any given design (though the EF can vary with design), with the interactions between a datum and a design, $(y, \xi)$, and the target quantities of interest, $\theta$, strictly through an inner product.

We then further provide a rigorous mathematical characterization of how certain degenerate behavior may arise when optimizing the EIG for EF-modeled experiments, including sufficient conditions for when the EIG is independent of $\xi$, monotonic in a (known) function of $\xi$, and when it is convex for any convex set of possible designs.

Finally, we empirically demonstrate the practical benefits of our singly-nested estimators for the EIG against widely-used nested estimator baselines. We show the desirable empirical convergence rates of our singly-nested estimators for a problem of increasing dimensionality, as well as the large performance improvements over the baselines with matched sample budgets when applied to a variety of BED problems.

## 2. Background

### 2.1. Exponential Family

We briefly review properties of Exponential Family (EF) distributions that will be central to our analysis.

**Definition 2.1.** [EF Form] An EF is a set of distributions

[1]Department of Statistics, University of Oxford. Correspondence to: Rik Knowles <knowles@stats.ox.ac.uk>.

*Proceedings of the $43^{rd}$ International Conference on Machine Learning*, Seoul, South Korea. PMLR 306, 2026. Copyright 2026 by the author(s).

that each take the form

$$\pi(\theta) = h(\theta) \exp\left(\lambda^\top T(\theta) - \Psi(\lambda)\right), \qquad (1)$$

where $h(\theta)$ is the 'base measure', $\lambda$ is the 'natural parameter' that indexes different elements of the EF family, $T(\theta)$ is the 'sufficient statistic', and $\Psi(.)$ is the 'log-partition function':

$$\Psi(\lambda) = \log\left(\int h(\theta) \exp\left(\lambda^\top T(\theta)\right) d\theta\right). \qquad (2)$$

Two distributions belong to the *same* EF if they can both be expressed in the functional form of definition (2.1), with the *same* sufficient statistic and the *same* base measure (noting that the log-partition function is derived from these), but with potentially *different* natural parameters. Note that any given distribution can simultaneously belong to more than one EF (e.g,. we can always add dimensions to $T(\theta)$ and set the corresponding new dimensions of $\lambda$ to 0).

The following is a key property of EFs that we will use in our derivations.

**Theorem 2.2.** *[Mean Parameter Equivalence] The mean parameter $\mu$ defined as $\mu := \mathbb{E}_{\pi(\theta)} T(\theta)$ is equal to (Wainwright & Jordan, 2008),*

$$\mu = \nabla_\lambda \Psi(\lambda). \qquad (3)$$

*In addition, if the exponential-family representation is minimal, in the sense that there is no nonzero $a$ and constant $b$ such that $a^\top T(\theta) = b$ almost surely under the base measure, then $\lambda \mapsto \mu$ is injective. For many common families, this mapping is available in closed form (Wainwright & Jordan, 2008).*

In this work, we say a prior is *conjugate* to a likelihood if the posterior belongs to the same EF as the prior and admits a closed-form update of its natural parameters.

### 2.2. Expected Information Gain

In the context of BED, one typically seeks experimental designs that, in expectation, are maximally informative. This notion gives rise to the Expected Information Gain (EIG) (Lindley, 1956; 1972; Bernardo, 1979), which has been successfully utilized as an objective function in a variety of contexts (Rainforth et al., 2024; Huan et al., 2024). Formally, the EIG can be written as the expectation with respect to the current marginal distribution over experiment outcomes, $p(y \mid d)$, of the KL divergence between the posterior and prior over our target quantities of interest $\theta$, with the former obtained after conditioning on a chosen experimental design setup $d$, and an outcome $y$,

$$\text{EIG}(\xi) \triangleq \mathbb{E}_{p(y|\xi)}[\text{KL}(p(\theta|y,\xi)\|p(\theta))], \qquad (4)$$

$$= \mathbb{E}_{p(\theta)p(y|\theta,\xi)}\left[\log p(\theta \mid y, \xi) - \log p(\theta)\right], \qquad (5)$$

$$= \mathbb{E}_{p(\theta)p(y|\theta,\xi)}\left[\log p(y \mid \theta, \xi) - \log p(y \mid \xi)\right], \qquad (6)$$

where the final line comes from a simple application of Bayes' theorem and can be interpreted as the expected reduction in predictive uncertainty from knowing $\theta$.

The EIG is a doubly-intractable quantity, and hence its estimation is expensive. Several Monte Carlo estimators have been proposed, the simplest of which is the *Nested Monte Carlo* (NMC) estimator (Rainforth et al., 2018),

$$\text{EIG}(\xi) \approx \frac{1}{N}\sum_{n=1}^{N}\left[\log p(y_n \mid \theta_n, \xi) - \log \hat{p}(y_n|\xi)\right], \quad (7)$$

where $\hat{p}(y_n|\xi) = \frac{1}{M}\sum_{m=1}^{M} p(y_n \mid \theta'_{n,m})$, $(\theta_n, y_n) \sim p(\theta)\,p(y \mid \theta, \xi)$, and $\theta'_{n,m} \overset{\text{i.i.d.}}{\sim} p(\theta)$. NMC is *consistent* as $N, M \to \infty$, but has $O(NM)$ cost and is biased for finite $M$ (Rainforth et al., 2018). A related estimator is the PCE estimator (Foster et al., 2021), where we include the additional outer theta sample inside the inner marginal likelihood estimate. This often has lower variance than NMC but is still biased, and maintains the $O(NM)$ cost.

## 3. Singly-Intractable $\text{EIG}(\xi)$ and $\nabla_\xi \text{EIG}(\xi)$ for EF Posteriors

The doubly intractable nature of the EIG is often a computational bottleneck to the application of BED. We now show though that, if we have an EF posterior over the parameter of interest whose dependence on designs and outcomes is through known natural parameters, then we can obtain a singly-intractable expression for the EIG.

**Theorem 3.1.** *Assume that there exists some EF such that the posterior, $p(\theta|y,\xi)$, always belongs to this EF regardless of $(y, \xi)$. That is, our posterior takes the functional form*

$$p(\theta|y,\xi) = h(\theta) \exp\left(\langle\lambda(y,\xi),\ T_p(\theta)\rangle - \Psi_p(\lambda(y,\xi))\right) \quad (8)$$

*with base measure, $h(\theta)$, sufficient statistic, $T_p(\theta)$, natural parameters that depend on the data and designs through a given function, $\lambda(y, \xi)$, and log-partition function $\Psi_p$ defined as per (2). Further, let $p(y|\xi)$ denote a marginal predictive distribution over data that yields a design-independent prior, i.e. $\int p(y|\xi)p(\theta|y,\xi)dy$ is constant in $\xi$.[1]*

*Then for any $\lambda_0 \in \mathbb{R}^d$ lying in the domain of the natural parameter, the resulting EIG of $p(\theta|y,\xi)p(y|\xi)$ is*

$$\text{EIG}(\xi) \overset{c}{=} \mathbb{E}_{p(y|\xi)}\Big[\left(\lambda(y,\xi) - \lambda_0\right)^T \mu(y,\xi) \\ - \Psi_p\big(\lambda(y,\xi)\big)\Big] \qquad (9)$$

---

[1]Note that, while our proof directly uses this property, the prior being independent of the design is already needed to be a true BED formulation, so this assumption does not restrict our setup.

where $\stackrel{c}{=}$ denotes equality up to constants in $\xi$ and $\mu(y, \xi) = \mathbb{E}_{p(\theta|y,\xi)}[T_p(\theta)]$ is the mean parameter.

Furthermore, assuming the existence of some continuously differentiable reparameterization function $g(\varepsilon, \theta, \xi)$, such that for some parameterless noise variable $\varepsilon \sim p(\varepsilon)$, we have $y = g(\epsilon, \theta, \xi) \sim p(y|\theta, \xi)$, then, for any $\lambda_0$, the gradient of the EIG can be expressed as,

$$\nabla_\xi \operatorname{EIG}(\xi) = \mathbb{E}_{p(\theta)p(\varepsilon)} \left[ (\lambda(y, \xi) - \lambda_0)^T \nabla_\xi \mu(y, \xi) \right]. \tag{10}$$

Proof. Let $q(\theta)$ denote any fixed distribution on $\theta$ with the same support as $p(\theta|y, \xi)$. Using this $q(\theta)$ and the fact that the prior $p(\theta)$ is independent of the design, we can obtain a useful rearrangement of the EIG (which holds regardless of whether the posteriors are EF), starting with Equation 5,

$$\begin{aligned}
\operatorname{EIG}(\xi) &= \operatorname{H}[p(\theta)] - \mathbb{E}_{p(y|\xi)} \operatorname{H}[p(\theta|y, \xi)] \\
&\stackrel{c}{=} -\mathbb{E}_{p(y|\xi)} \operatorname{H}[p(\theta|y, \xi)] \\
&\stackrel{c}{=} -\mathbb{E}_{p(\theta)}[\log q(\theta)] - \mathbb{E}_{p(y|\xi)} \operatorname{H}[p(\theta|y, \xi)] \\
&= \mathbb{E}_{p(y|\xi)}[\operatorname{KL}(p(\theta|y, \xi) \| q(\theta))] \tag{11}
\end{aligned}$$

We can now choose $q(\theta)$ to be in the same EF as the posterior, using $\lambda_0$ to denote its corresponding natural parameters. This choice now allows us to invoke the known formula for the KL-divergence between two distributions in the same EF (Nielsen & Nock, 2010). Using this, we then have

$$\begin{aligned}
\operatorname{EIG}(\xi) \stackrel{c}{=} \mathbb{E}_{p(y|\xi)} \big[ (\lambda(y, \xi) - \lambda_0)^T \mu \\
- \Psi(\lambda(y, \xi)) + \Psi(\lambda_0) \big].
\end{aligned}$$

Finally, we get the required form of (9) by noting that $\Psi(\lambda_0)$ is independent of $\xi$ and absorbing it into the constant.

For the gradient result, using the shorthands, $\lambda := \lambda(g(\varepsilon, \theta, \xi), \xi)$, $\Psi_1 := \Psi(g(\varepsilon, \theta, \xi), \xi)$, and $\mu := \mu(g(\varepsilon, \theta, \xi), \xi)$, we have,

$$\begin{aligned}
\nabla_\xi \operatorname{EIG}(\xi) &= \nabla_\xi \mathbb{E}_{p(y|\xi)} \left[ \operatorname{KL}(p(\theta \mid y, \xi) \| q(\theta)) \right] \\
&= \nabla_\xi \mathbb{E}_{p(\theta) p(y|\theta; \xi)} \left[ (\lambda - \lambda_0)^\top \mu - \Psi_1 \right] \\
&= \mathbb{E}_{p(\theta) p(\varepsilon)} \nabla_\xi \left[ (\lambda - \lambda_0)^\top \mu - \Psi_1 \right] \\
&= \mathbb{E}_{p(\theta) p(\varepsilon)} \big[ \mu^\top \nabla_\xi \lambda + (\lambda - \lambda_0)^\top \nabla_\xi \mu \\
&\qquad\qquad - (\nabla_\lambda \Psi)^\top \nabla_\xi \lambda \big] \\
&= \mathbb{E}_{p(\theta) p(\varepsilon)} \left[ (\lambda - \lambda_0)^\top \nabla_\xi \mu \right].
\end{aligned}$$

where the required result (10) then follows directly from Theorem 2.2. □

Theorem 3.1 shows that the posterior always belonging to the same EF is sufficient to obtain a singly-intractable

EIG. Importantly, we have not made any assumptions on conjugacy, or even any direct assumptions on the prior or likelihood at all (other than they form a valid joint for BED). Note, though, that always being in the same EF itself implies that the posterior only depends on $(y, \xi)$ through the natural parameters. This itself restricts how $\xi$ and $\theta$ can interact in the posterior density. Namely, we have

$$\log p(\theta|y, \xi) = \langle \lambda(y, \xi), T_p(\theta) \rangle + \log h(\theta) - \Psi_p(\lambda(y, \xi))$$

so our assumption implies that the log posterior density is linear in functions that each depend only on one of $\theta$ or $(y, \xi)$ but not both simultaneously.

From Theorem 3.1, we can straightforwardly derive singly intractable estimates for the EIG and its reparameterized derivative by making simple Monte Carlo estimators of (9) and (10) respectively. For the former we need only sample $y$ using $y \sim p(y|\xi)$, while in the latter we are also sampling $\theta$ to exploit the reparameterization trick.

Note that if $y|\xi$ is discrete, bounded and the prior predictive density $p(y|\xi)$ is known analytically, the EIG and its derivative are both fully analytic, as we can explicitly evaluate the outer expectation. Where the reparameterization trick (Kingma & Welling, 2013), or variants like the implicit reparameterization trick (Figurnov et al., 2018), cannot be applied, we may resort to the REINFORCE estimator (Williams, 1992) (see Appendix A for the resulting estimator).

## 4. Obtaining EF Posteriors

In general, we do not directly specify the posterior form, so it is not immediately clear under what model setups the conditions for Theorem 3.1 to hold will actually occur. The following result now reveals under what models we can utilize our singly intractable EIG estimators by providing a condition on the likelihood to achieve posteriors of the required EF form.

**Theorem 4.1.** If a model's likelihood function can be written in the form,

$$p(y \mid \theta, \xi) = h_\ell(y, \xi) \exp\left( \langle \lambda(y, \xi), T_p(\theta) \rangle - \Psi_\ell(T_p(\theta)) \right) \tag{12}$$

and further yields a log-partition function,

$$\Psi_\ell(T_p(\theta)) = \log \int h_\ell(y, \xi) e^{\langle \lambda(y,\xi), T_p(\theta) \rangle} \, dy,$$

that is itself independent of $\xi$ for all possible $\theta$, then the resulting posterior takes the required EF form of Theorem 3.1 with the corresponding reference measure

$$h(\theta) = p(\theta) \exp\left( -\Psi_\ell(T_p(\theta)) \right), \tag{13}$$

where $p(\theta)$ is the corresponding prior we use (which is unrestricted other than in being independent of $\xi$).

*Proof.* Using Bayes' rule we have

$$p(\theta|y,\xi) \propto p(\theta)p(y|\theta,\xi)$$
$$\propto h(\theta) \exp\left(\langle \lambda(y,\xi),\, T_p(\theta)\rangle\right),$$

which yields the required EF form from Equation (8) once normalised, noting that the log partition function, $\Psi_\ell(T_p(\theta))$, being independent of $\xi$ ensures that $h(\theta)$ is also independent of $\xi$. $\quad\square$

Further, it can be shown that this form is also necessary to obtain an EF posterior of the form in Theorem 3.1 (see Appendix B).

We see from this result that the requirement on our likelihood is somewhat analogous to that we previously had on our posterior: we must be able to write the log-likelihood density as a linear mapping of functions that each themselves depend on only one of $\theta$ or $(y,\xi)$. In other words, our likelihood forms an exponential family with base measure $h_\ell(y,\xi)$, natural parameters $T_p(\theta)$, sufficient statistics $\lambda(y,\xi)$, and log-partition function $\Psi_\ell(T_p(\theta))$. Unlike the posterior, this EF can *change* depending on the design, but $\theta$ is only allowed to index members of the EF (through its own natural parameter mapping $T_p(\theta)$).

Note here that we are not restricted in the forms we choose for $h_\ell(y,\xi)$, $\lambda(y,\xi)$, and $T_p(\theta)$ when constructing a model, provided the requirement of the log partition being independent of $\xi$ is satisfied. The allowed model class is thus actually very flexible as we can use a high dimensional set of natural parameters that then induces complex non-linear dependencies between $y$, $\xi$, and $\theta$, analogously to kernelisation of linear methods (Hastie et al., 2009).

In general though, most given likelihoods cannot be written in the required form using a *finite* number of natural parameters. For example, in a Bayesian neural network there is a complex non-linear relationship between the weights and inputs/outputs that cannot be represented in closed form using an inner product on separate mappings of each. On the other hand, if one uses a Bayesian neural network where only the last layer weights are stochastic, then it generally will be possible to express the likelihood in the required form.

### 4.1. Conjugacy

In our previous results, we have made no assumptions about our model being conjugate. However, it turns out that we can interpret models that have our desired EF properties as satisfying an underlying conjugacy by reformulating the given posterior form to one with an expanded set of natural parameters and sufficient statistics. Namely, if we consider our natural parameters to be $(\lambda, n-1)$ instead of just $\lambda$ where $n$ is a number of observations, and our sufficient statistics to be $(T_p(\theta), -\Psi_\ell(T_p(\theta)))$ instead of just $T_p(\theta)$,

then we can see that this forms an expanded EF that contains both the prior and the posterior. Namely, the prior has the particular parameter configuration $\lambda = 0$ and $n = 0$, while the posterior has $\lambda = \lambda(y,\xi)$ and $n = 1$.

The importance of this becomes apparent when we consider the sequential experimental design setting as it implies that the posterior remains in this same expanded EF under repeated observation. Namely, in the standard setting where data is i.i.d. given the parameters and designs, then the following result holds which allows us to straightforwardly apply our estimators in the sequential setting.

**Lemma 4.2.** *Given $n$ observations $y_1, \ldots, y_n$ that are conditionally independent of each other given $\theta$ and a corresponding set of designs $\xi_1, \ldots, \xi_n$ and each generated according to an identical likelihood taking the form Equation* (12)*, then the corresponding posterior is given by*

$$p(\theta \mid y_{1:n}, \xi_{1:n}) = h(\theta)\exp\big(\langle \lambda_\Sigma,\, T_p(\theta)\rangle$$
$$- (n-1)\,\Psi_\ell(T_p(\theta)) - \Psi_p(\lambda_\Sigma, n)\big).$$

*where $\lambda_\Sigma = \sum_{i=1}^n \lambda(y_i, \xi_i)$ and the reference measure $h$ is derived from the prior as per Equation* (13)

*Proof.* Using the assumed conditional independence of observations and Bayes' rule,

$$p(\theta \mid y_{1:n}, \xi_{1:n}) \propto p(\theta) \prod_{i=1}^n p(y_i \mid \theta, \xi_i). \qquad (14)$$

We further have,

$$p(\theta) = h(\theta)\exp\left(\langle 0, T_p(\theta)\rangle + \Psi_\ell(T_p(\theta))\right),$$
$$p(y_i|\theta,\xi_i) = h_\ell(y_i,\xi_i)\exp\left(\langle \lambda(y_i,\xi_i), T_p(\theta)\rangle - \Psi_\ell(T_p(\theta))\right).$$

The result now follows straightforwardly from substituting the above into Equation (14), dropping the $h_\ell(y_i, \xi_i)$ terms as these are independent of $\theta$, and renormalising. $\quad\square$

This result further implies that we can only achieve our singly intractable EIG estimators when there is an underlying conjugacy present. However, this in itself is a weaker requirement than it first seems: we have shown that likelihoods of the required form are conjugate to *any* choice of prior, with the EF of this conjugacy changing with both the prior and the likelihood we choose. Thus, we are not restricted to only using well-established known conjugate pairings, but have instead derived the, potentially highly complex, EF that our posteriors will remain in.

## 5. Limitations and Degeneracy

Although not necessarily the case, as shown in Section 7, some naive EF-modeled setups are degenerate optimization problems in the context of BED. For example, the EIG($\xi$)

may be independent of $\xi$, or may be monotonic/convex in some analytic and known function of $\xi$. We now carefully characterise these degeneracies through a series of results.

**Removing Design Dependency via a Change of Variable:** Because the EIG is invariant to invertible transformations of $\theta$ or $y$, if we can remove the influence of the design through such an invertible transformation then our experiment setup was not meaningful, whether we are working with an EF model or not. For example, if the likelihood for $Y$, is given by $p(y|\xi,\theta)$ and there exists an invertible transformation $Z = Z(Y, \xi)$, such that the new likelihood on $Z$, $p(z|\theta)$ is independent of $\xi$, then the EIG $:= I(\theta; Y) = I(\theta; Z)$ is independent of the design. The proof follows from Theorem 2.8.1 in (Cover, 1999). **Example:** Take $Y \sim \mathrm{Exp}(B(\xi)\theta)$, then we can let $X = B(\xi)Y$ so that $X \sim \mathrm{Exp}(\theta)$. Since the transformation is one-to-one and the distribution of $X$ does not depend on $\xi$, the EIG is independent of $\xi$.

**Monotonicity in the exposure variable:** We next show that for a special case of the likelihood form presented in Theorem 4.1 then EIG becomes monotonic in mapping of the designs that we call the exposure function, $c(\xi)$. Namely, as formalised in the following result, if the terms inside the exponential can be broken down into the difference of inner products of functions that themselves only depend on one of $y$, $\theta$, and $\xi$, then the EIG will become monotonic in the corresponding function of $\xi$, which is the exposure function $c(\xi)$.

**Theorem 5.1.** *Assume a likelihood of the form*

$$p(y \mid \theta, \xi) = h(y, \xi)\exp\{\langle S(y), \eta(\theta)\rangle - \langle c(\xi), A(\eta(\theta))\rangle\},$$

*where $c(\xi) \in \mathbb{R}_{\geq 0}^m$ is a continuous vector-valued function, referred to as the exposure function, and $S(y)$ is a sufficient statistic of $y$ such $I(\theta; y) = I(\theta; S(y))$ (which we assume is sufficiently regular for $S(y)$ to admit a characteristic function). Then the $\mathrm{EIG}(\xi)$ is element-wise monotonic in $c(\xi)$.*

*Proof.* See Appendix C. □

This likelihood structure implies the design reweighs the curvature contributions $\nabla_\eta^2 A_j(\eta)$, seen via the Hessian of the negative log-likelihood, $-\nabla_\eta^2 \log p(y|\theta, \xi) = \nabla_\eta^2 \langle c(\xi), A_\eta(\eta)\rangle = \sum_{j=1}^m c_j(\xi)\nabla_\eta^2 A_j(\eta)$. Thus increasing each $c_j(\xi) \geq 0$, typically makes the likelihood more sharply peaked in $\eta$, implying more information per observation, and a more concentrated posterior.

We refer to $c(\xi)$ as the 'exposure function', because it acts like an effective sample size, or exposure variable, which can be seen in the following example, **Example:** Assume a Poisson likelihood $p(y|\xi, \theta) = (1/y!)\exp(\langle y, \eta(\theta)\rangle - \exp(\eta))$ with rate $c(\xi)\theta$, and natural parameter, $\eta(\theta) = \log(c(\xi)\theta))$. By redefining a new natural parameter, $\tilde{\eta}(\theta) = \log(\theta)$, we instead can write the likelihood as, $p(y|\xi, \theta) = (c(\xi)^y/y!)\exp(\langle y, \tilde{\eta}(\theta)\rangle - c(\xi)\exp(\tilde{\eta}))$. Hence the $\mathrm{EIG}(\xi)$ is monotonic in $c(\xi)$.

Although the EIG is monotonic in the exposure function, in a real-world setting, there is likely an increased cost associated with increasing the exposure function, e.g,, a cost proportional to the time spent observing a natural phenomenon. Thus the true objective function would be a combination of the EIG and a cost function, and this combination will not generally be monotonic in the exposure function.

**Convexity of the EIG under Affine Posterior Means:** Finally, we show a sufficient condition for the EIG to be convex in the designs, such that the optimal design always occurs at an extremum of the design space.

**Theorem 5.2.** *Let $\Xi$ be a convex design space. Suppose that, for every $(y, \xi)$, the posterior $p(\theta \mid y, \xi)$ belongs to a common EF as per Theorem 3.1 with mean parameter*

$$\mu_{\mathrm{post}}(y; \xi) := \mathbb{E}_{p(\theta|y,\xi)}[T(\theta)].$$

*Assume that the prior $p(\theta)$ is independent of $\xi$, and that the likelihood admits a reparameterization*

$$y = g(\theta, \varepsilon, \xi), \qquad \theta \sim p(\theta), \qquad \varepsilon \sim p(\varepsilon),$$

*where $p(\varepsilon)$ is independent of both $\theta$ and $\xi$, and $g(\theta, \varepsilon, \xi) \sim p(y \mid \theta, \xi)$. If the map*

$$\xi \mapsto \mu_{\mathrm{post}}\big(g(\theta, \varepsilon, \xi); \xi\big)$$

*is affine on $\Xi$ for $p(\theta)p(\varepsilon)$-almost every $(\theta, \varepsilon)$, then $\mathrm{EIG}(\xi)$ is convex on $\Xi$.*

*Proof.* Let $q(\theta)$ be any fixed distribution in the same exponential family as the posterior, with natural parameter $\lambda_0$ and mean parameter $\mu_0$. Further, let $\Psi^*$ denote the convex conjugate of $\Psi$,

$$\Psi^*(\mu) = \sup_{\lambda \in \mathrm{dom}(\Psi)} \{\langle \lambda, \mu\rangle - \Psi(\lambda)\}.$$

For two members of the same regular exponential family, the KL divergence has the mean-parameter Bregman representation (Nielsen (2023), see Appendix D),

$$\mathrm{KL}\big(p(\theta \mid y, \xi) \,\|\, q(\theta)\big) = D_{\Psi^*}\left(\mu_{\mathrm{post}}(y; \xi) \,\|\, \mu_0\right),$$

where

$$D_{\Psi^*}(\mu\|\mu_0) = \Psi^*(\mu) - \Psi^*(\mu_0) - \langle \nabla\Psi^*(\mu_0), \mu - \mu_0\rangle.$$

Since $\Psi^*$ is convex, $D_{\Psi^*}(\mu\|\mu_0)$ is convex in its first argument $\mu$. Now taking an expectation over $p(y|\xi)$ and using our reparameterisation we have using the above,

$$\mathrm{EIG}(\xi) \overset{c}{=} \mathbb{E}_{p(y|\xi)}\left[\mathrm{KL}\big(p(\theta \mid y, \xi)\,\|\, q(\theta)\big)\right],$$
$$= \mathbb{E}_{p(\theta)p(\varepsilon)}\left[D_{\Psi^*}\big(\mu_{\mathrm{post}}(g(\theta, \varepsilon, \xi); \xi) \,\|\, \mu_0\big)\right].$$

For any given $(\theta, \varepsilon)$, we have by assumption that $\xi \mapsto \mu_{\mathrm{post}}\big(g(\theta, \varepsilon, \xi); \xi\big)$ is affine on $\Xi$. Therefore the composition

$$\xi \mapsto D_{\Psi^*}\big(\mu_{\mathrm{post}}\big(g(\theta, \varepsilon, \xi); \xi\big) \,\|\, \mu_0\big)$$

is convex on $\Xi$. Finally, the expectation above is taken with respect to the fixed distribution $p(\theta)p(\varepsilon)$, which is independent of $\xi$, and nonnegative weighted averages of convex functions are convex. The $\mathrm{EIG}(\xi)$ is thus itself convex as it only differs from the expected Bregman divergence by a term that is constant in $\xi$. $\qquad\square$

Since we want to maximize $\mathrm{EIG}(\xi)$, experiments for which Theorem 5.2 applies can be viewed as degenerate in the sense that, over a convex design space, maximizing a convex objective will typically drive the optimum to the boundary, or to infinity if the design space is unbounded.

The condition in the theorem should be interpreted as a pathwise affine mapping under the likelihood reparameterization. In particular, it is not sufficient that, for each fixed observation $y$, the map $\xi \mapsto \mu_{\mathrm{post}}(y; \xi)$ is affine: convexity is preserved only when the reparameterized posterior mean parameter is affine in $\xi$ for almost every $(\theta, \varepsilon)$.

This condition is satisfied whenever the reparameterized posterior natural parameters depend affinely on $\xi$ and the EF mean map is affine in those natural parameters. For Gaussian EF posteriors this is often easy to verify, since the log-partition is quadratic and the mean-parameter map is affine in the natural parameters. However, Gaussianity alone is not enough: the design dependence must remain affine after substituting the likelihood reparameterization $y = g(\theta, \varepsilon, \xi)$.

## 6. Related work

Traditional approaches to computing the EIG, beyond nested estimators, include Laplace approximations for posterior estimation (Long et al., 2013; Lewi et al., 2009; Cavagnaro et al., 2010; Long, 2022). Whereas Laplace-based posterior approximation offer computational speedups, they typically introduce substantial bias (Rainforth et al., 2024).

Other works have aimed to amortize marginal likelihood and posterior estimation by introducing variational approximations. These methods replace the nested expectation with a variational bound, which can be optimized directly, in lieu of the EIG (Foster et al., 2019; 2020; 2021). We note that despite the fact variational distributions are often within an EF, the outer expectation cannot be collapsed as it is not with respect to the variational approximation. When our estimators are applicable, using them directly for optimization is much more efficient than optimizing variational bounds, as we do not have to perform a joint optimization over variational parameters alongside the design.

One might ask if there are useful alternatives to variational bounds when we assume a variational posterior, but not necessarily our true posterior, is in the EF. Indeed, one could target the expected KL divergence between the next-step and current variational posteriors, as has recently been analyzed in the context of continual-learning (Khan, 2025), and uncertainty quantification (Bickford Smith et al., 2025) (see Appendix I). For EF variational posteriors, we could utilize analogous results to Theorem 3.1 to ensure this objective is singly intractable, for fixed variational approximations. However, in the context of BED, this offers no computational benefits over using the Barber-Agakov variational bound (Barber & Agakov, 2004; Foster et al., 2019). It also opens up the potential for pathologies, choosing points where the approximation is poor rather than the true EIG is high.

More recent work has derived a model-agnostic, singly-nested expression for the gradient of the EIG, that is also unbiased *if* exact samples from the posterior can be drawn (Ao & Li, 2023). However, in general, exact sampling from the posterior is itself intractable, and so in practice, the expressions derived are still either biased or doubly intractable to compute. Multi-level Monte Carlo methods have also been applied to estimating the EIG and its gradients (Goda et al., 2022). These use infinite series expansions to derive unbiased estimates of the EIG, but this typically comes at the expense of increased variance and the estimators are still nested, in the sense that we need multiple inner samples for each term in our outer estimator.

To the best of our knowledge, exploiting the analytic or singly-intractable forms for the EIG has been restricted to the case of Gaussians/GPs (Houlsby et al., 2011; Takeno et al., 2025). Our work is a significant generalization, showing that in any case where we have an EF posterior with design-dependency through the natural parameters, the EIG, and its gradient, are singly-intractable.

## 7. Experiments

We compare our singly-intractable estimator (**SI**) against two baselines for estimating and optimizing the EIG: the **NMC** estimator in Equation 7 and the corresponding **PCE** estimator, where the outer sample of $\theta$ is included in the inner marginal likelihood estimate.

We first study empirical convergence rates for **NMC**, **PCE**, and **SI** under a fixed total sample budget. We then evaluate the estimators in three BED tasks: 1) learning about the rate parameter of a **Poisson** process 2) learning about the scale matrix of a **Wishart** distribution and, 3) active **GP** regression using expected predictive information gain (EPIG).

**Convergence Rates** In Figure 1, we analyze the empirical

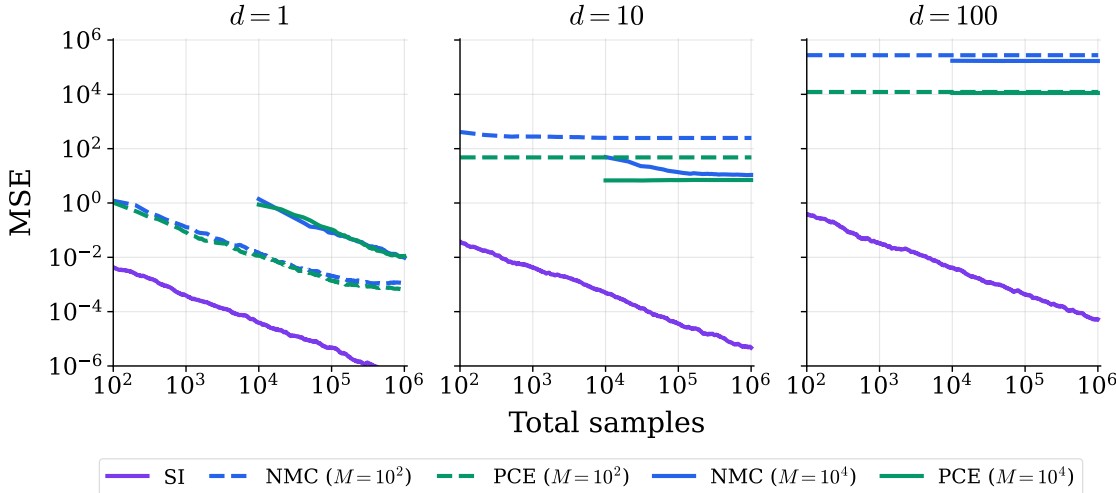

*Figure 1.* Empirical convergence of **NMC**, **PCE**, and **SI** using an equivalent number of *total* samples. For **NMC**, and **PCE** we fix $M$ and let $N$ increase, following (Rainforth et al., 2018). We plot the Mean Square Error (MSE) and the standard error (shaded), using 200 independent runs. We calculated the ground truth EIG using the analytic formula. We fix $\xi = 1.5$, and use the prior-likelihood set-up described in Appendix G.1.

convergence rates of **NMC**, **PCE**, and **SI** using an equivalent number of *total* samples for a snapshot of a Gaussian linear regression experiment in 1-D, 10-D and 100-D. This choice of experiment allows us to compare against an analytic ground truth EIG. We clearly see that the **SI** is significantly more efficient than the baselines and avoids the bias they introduce for finite $M$ (as shown by their flattening MSEs). The benefit of using the **SI** estimator increases with dimensionality. Indeed, in 100-D **both NMC** and **PCE** fail to reach an average Mean Square Error (MSE) lower than $10^4$, whereas the **SI** MSE reaches $10^{-4}$. For all dimensions, the **SI** MSE is centered around $-1$ gradient line, in agreement with the theoretical Monte Carlo convergence rate of $O(1/T)$, where $T$ is the total number of samples. Furthermore, we see its MSE at any fixed sample size increases linearly with the dimensionality, and so can scale to very high dimensions (see Appendix H for its performance in 1000-D) For more details on the likelihood and prior used see Appendix G.1.

**Poisson** In this experiment, we consider observations that follow a Poisson distribution whose rate is proportional to an underlying parameter $\theta$ scaled by an exposure variable. A basic property of Poisson processes is that the expected event count increases linearly with exposure, which has been empirically validated across a wide range of scientific settings, from radioactive decay—where exposure corresponds to observation time (Rutherford et al., 1910)—to cell counts in fluid suspensions, in which the rate increases proportionally with the volume (Student, 1907). In our experiments, we take the exposure function to be the area of the shape defined by four two-dimensional co-ordinates.

Specifically, we take $y|\xi, \theta \sim \text{Poisson}(\theta \times \text{Area}(\xi))$,

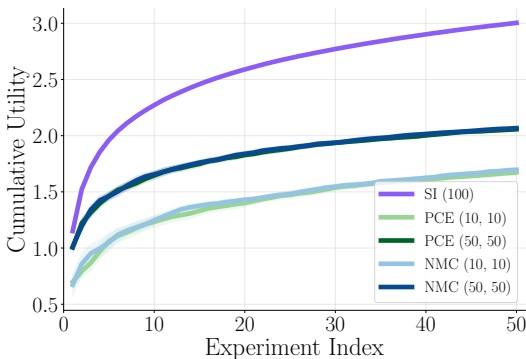

*Figure 2.* Poisson Experiment Cumulative Utility. Cumulative utility across experiment iterations when optimizing the EIG using the **SI** estimator with 100 outer samples, and the **NMC** and **PCE** estimator with 10 inner and 10 outer samples, and 50 inner and 50 outer samples. **NMC** and **PCE** overlap for both sample size sets.

where $\text{Area}(\xi)$ is the area of the rhombus defined by $\xi = \{\xi_1, \xi_2, \xi_3, \xi_4\} \in \mathbb{R}^{4 \times 2}$. This represents an experiment where we want to count a natural phenomenon, e.g, counts for individuals of a specific animal species, and we have control over the region of space observed.

In this experiment, the EIG is monotonic increasing with respect to the exposure parameter, as the likelihood satisfies Theorem 5.1. However, it is natural to assume that an increasing cost is associated with increasing the magnitude of the exposure parameter. Hence, we add a cost function, $C(\xi)$, to our overall utility, $U(\xi) = \text{EIG}(\xi) - \gamma C(\xi)$, and ensure an interior optimum. We take $C(\xi) = \sum_i \|\xi_i\|_2$, and $\gamma = 0.05$. For the prior, we take a Gamma conjugate prior. See Appendix G.2 for more details.

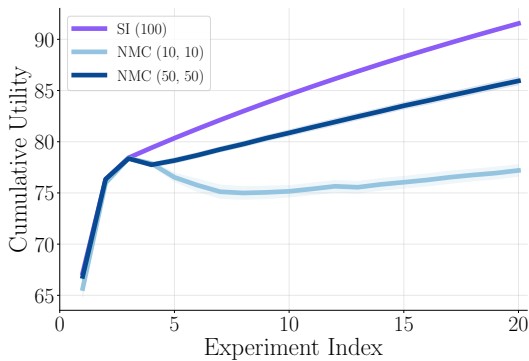

*Figure 3.* Wishart Experiment Cumulative Utility. Cumulative utility across experiment iterations when optimizing the EIG using the **SI** estimator with 100 outer samples, and the **NMC** with $(10, 10)$, and $(50, 50)$, outer, inner sample size pairings. **NMC** and **PCE** have very similar results and so we only plot **NMC**.

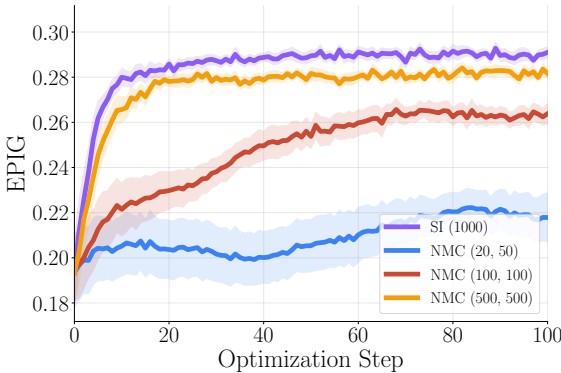

*Figure 4.* The mean and standard errors of (ground-truth) EPIG of a batch of 5 inputs across optimization for GP experiment when using the singly-intractable form, or the standard Nested Monte Carlo form (NMC) with differing numbers of inner and outer samples, denoted by $K$ and $N$ respectively. We average over 30 seeds. At each optimization step, we take the current design $\xi^t$, and calculate the ground truth EPIG value using NMC with $10,000$ outer and inner samples, which we found to be stable.

In Figure 2 we see, using **SI** for optimization leads to a substantial improvement in the cumulative utility (i.e. cumulative sum of utilities $U(\xi)$ achieved at each iteration) across all experiment iterations. Indeed, even with a total of $\sim 25$ times more total samples, the final cumulative utility for **NMC** and **PCE** is almost half that of **SI**.

**Wishart** We now consider modeling observations that sample covariance matrices computed from Gaussian data, with the goal of learning the true underlying covariance matrix.

Let $x_1, \ldots, x_{n(\xi)} \overset{\text{i.i.d.}}{\sim} \mathcal{N}_p(0, \Sigma)$, be $n(\xi) = \xi$ independent draws from a zero-mean $p$-dimensional multivariate normal distribution with unknown covariance matrix, $\Sigma$. The (unnormalized) sample covariance matrix is then given by, $y = \sum_{i=1}^{n(\xi)} x_i x_i^\top$. Under this generative model, the random matrix, $Y$, follows a Wishart distribution, $Y \mid \Sigma, \xi \sim \text{Wishart}(n(\xi), \Sigma)$, where $n(\xi)$ is called the degrees of freedom. We let $p = 10$, and place an Inverse-Wishart prior over $\Sigma$.

This likelihood satisfies Theorem 5.1. However, it is natural to assume an increasing cost with increasing the degrees of freedom parameter, $n(\xi)$. We therefore take our utility function to be $U(\xi) = \text{EIG}(\xi) - \gamma\, C(\xi) + 1$, where $C(\xi) = n(\xi)$, and $\gamma = 0.1$ (the benign addition of 1 is chosen for plotting aesthetics). See Appendix G.3 for more details.

In Figure 3, we again see, despite having over 25 times the sampling budget, **NMC** (and **PCE**) achieve a significantly lower cumulative utility.

**GP** For the final experiment, we investigate active GPR by maximizing the EPIG. Namely, we select a single *batch* of five input locations, $\mathbf{X} = \{x_1, \ldots, x_5\}$, at which to evaluate the GP and observe the corresponding outputs $\mathbf{y} = \{y_1, \ldots, y_5\}$, by optimizing, (Smith et al., 2023),

$$\text{EPIG}(\mathbf{X}) = \mathbb{E}_{p_*(x_*)}[I(y; y_* \mid \mathbf{X}, x_*)]. \quad (15)$$

We consider the likelihood $p(y \mid f(x)) = \mathcal{N}(y \mid f(x), \sigma_n^2)$, and place a GP prior over the latent function, $f \sim \mathcal{GP}(0, k_\phi(\cdot, \cdot))$, where $k_\phi$ is an RBF kernel parameterized by hyperparameters $\phi$ (Rasmussen & Williams, 2006). Further experimental details are provided in Appendix G.4.

Given our model choice, for a candidate input location $x_*$ with associated output $y_*$, the joint predictive distribution of $\mathbf{y}$ and $y_*$, conditioned on $\mathbf{X}$ and $x_*$, is Gaussian and admits a closed-form expression for the mutual information $I(\mathbf{y}; y_* \mid \mathbf{X}, x_*)$. Therefore, in this case, $\text{EPIG}(\mathbf{X})$ can be reduced to a singly-intractable quantity. We denote the corresponding singly-intractable estimator, **SI**. We denote the estimator in which we also perform a nested estimation of the mutual information, **NMC**. For more details on the resulting estimators, see Appendix E.1.

Although EPIG has been previously used for active GPR (Takeno et al., 2025), crucially, a direct comparison of using the singly-intractable estimator, **SI**, versus the standard nested estimator, **NMC**, is lacking in the literature.

Figure 5 shows the substantial qualitative difference between the designs chosen by **SI** and **NMC**. In Figure 4, meanwhile, we see optimization trajectories of EPIG, using **SI**, versus **NMC**. For the singly-intractable estimator, we use 1000 outer samples per optimization step. We see that when using a sample budget comparable to that of **SI**, **NMC** fails to successfully optimize EPIG. It requires 500 outer samples and 500 inner samples, totaling 250 times more Monte Carlo samples than the SI estimator, to achieve a comparable performance, showing the scalability of **SI**, and the shortcoming of the standard baselines.

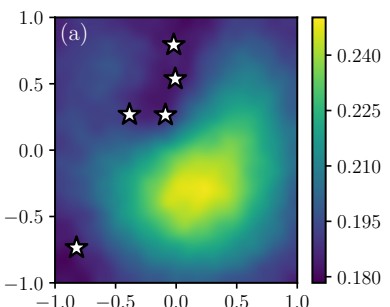 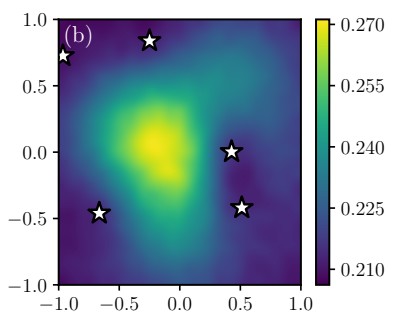 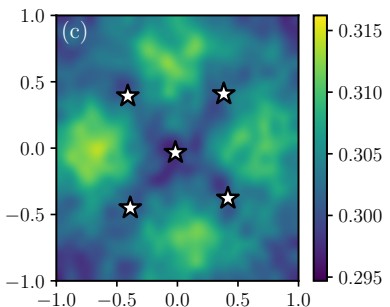

*Figure 5.* Ground-truth EPIG values calculated at 1000 equally spaced locations over a $[-1, 1]^2$ grid, *after* conditioning on 5 input locations, given by (a) Randomly chosen points. (b) Batch optimized using **NMC** with $N = 20$ outer and $K = 50$ inner samples. (c) Batch optimized using **SI** with $N = 1000$ outer samples. Ground truth EPIG values calculated using NMC with $10^7$ total samples.

## 8. Conclusion and Future Work

This work serves to answer a long-standing open question in the BED community; besides when our likelihood and prior are Gaussian, is there a more general class of models that admit tractable EIGs?

We prove that a posterior distribution belonging to the exponential family (EF) and depending on the experimental design only through known natural parameters is a sufficient condition to collapse the expected information gain (EIG) from a doubly intractable to a singly intractable expression. In such cases, we provide singly-intractable and unbiased estimators for the EIG and its (reparameterized) gradient which converge linearly, as seen in Figure 1.

However, Section 5 shows a naive EF-model choice can lead to degenerate behavior when optimizing the EIG. Although, in practice, even if the EIG itself is monotonic/convex, our overall objective function may not be, and our experiments clearly show that more *accurate* estimators of the EIG are useful when optimizing a utility, to correctly weight the value of information against a cost function, as seen in Figures 2 and 3.

An important question is how our results may be leveraged if the assumptions in this paper do not hold exactly. In particular, if the condition in Theorem 4.1 does not hold directly, one could approximate the likelihood using a basis expansion, after which our singly-intractable estimators become applicable. While this would inevitably add to the cost of the estimator as the number of basis functions used grows, it may have a better compute-accuracy trade-off than PCE/NMC, particularly when the likelihood is close to its approximation.

Hypothetically, one might use EF variational approximations to the posterior and target the expected KL divergence between the current and next-step posterior. However, there are several shortcomings to this, and we do not see advantages over using BA bounds (Foster et al., 2019). See Appendix I for more details.

Finally, we see a potential to increase the efficiency of the standard NMC/PCE estimator by using an EF proposal to introduce a control variate. Specifically, we can add and subtract the analytic information gain between a proposal and some distribution in the same EF, to the true information gain. Under certain rearrangements, this is likely to reduce variance provided the proposal is close to the posterior.

## Acknowledgments

RK is supported by the Martingale Postgraduate Foundation. TR is supported by the UK EPSRC grant EP/Y037200/1. The authors would like to thank Sahel Iqbal for feedback on an earlier draft of the manuscript.

## Impact Statement

This paper presents work whose goal is to advance the field of Machine Learning. There are many potential societal consequences of our work, none which we feel must be specifically highlighted here.

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

## A. Singly-Intractable REINFORCE Estimator of the Gradient of the EIG

Assume the posterior over the parameter of interest is given by

$$p(\theta \mid y, \xi) = h(\theta) \exp\big(\lambda(y, \xi)^T T(\theta) - \Psi(\lambda(y, \xi))\big),$$

and take $\lambda_0$ to be any fixed element of the natural parameter space of the exponential family. Let $\mu(y, \xi)$ denote the posterior's mean parameter. The resulting REINFORCE Estimator of Gradient of the EIG, making use of EF properties as in 3.1, is

$$\nabla_\xi \operatorname{EIG}(\xi) = \mathbb{E}_{p(y|\xi)}\Big[\big(\nabla_\xi \log p(y|\xi)\big) g(y, \xi) + (\lambda(y, \xi) - \lambda_0)^T \nabla_\xi \mu(y, \xi)\Big],$$

where $g(y, \xi) = (\lambda(y, \xi) - \lambda_0)^T \mu(y, \xi) - \Psi(\lambda(y, \xi))$.

## B. Proof of Necessary Condition on Likelihood for EF Posterior

*Proof.* By Bayes' rule,

$$\log p(y \mid \theta, \xi) = \log p(\theta \mid y, \xi) - \log p(\theta) + \log p(y \mid \xi).$$

Substituting the posterior form yields

$$\log p(y \mid \theta, \xi) = \log h(\theta) - \log p(\theta) + \langle \lambda(y, \xi),\ T_p(\theta) \rangle + \log p(y \mid \xi) - \Psi_p\big(\lambda(y, \xi)\big) \tag{16}$$

$$= \langle \lambda(y, \xi),\ T_p(\theta) \rangle + \log h_\ell(y, \xi) + \text{const. in } y \text{ and } \xi, \tag{17}$$

where $\log h_\ell(y, \xi) = \log p(y \mid \xi) - \Psi_p(\lambda(y, \xi))$. Hence, the likelihood is given by,

$$p(y|\theta, \xi) = h_\ell(y, \xi) \exp\{\langle \lambda(y, \xi),\ T_p(\theta) \rangle - \Psi_l(T_p(\theta))\} \tag{18}$$

where $\Psi_l$ has been implicitly defined as the log partition function required to ensure that the likelihood is a well normalised distribution. Note that because the missing constant term is $\log p(\theta) - \log h(\theta)$, this further implies that the corresponding prior will be of the form

$$p(\theta) = h(\theta) \exp(\Psi_\ell(T_p(\theta))). \tag{19}$$

$\square$

### B.1. Proof of Conjugacy Class for Nth-step Posterior under Independent Observations

Using conditional independence of observations,

$$p(\theta \mid y_{1:n}, \xi_{1:n}) \propto p(\theta) \prod_{i=1}^{n} p(y_i \mid \theta, \xi_i), \tag{20}$$

Substituting in the prior form from Equation (19) and the likelihood form from Equation (18),

$$\propto h(\theta) \exp(\Psi_\ell(T_p(\theta))) \cdot \prod_{i=1}^{n} h_\ell(y_i, \xi_i) \exp(\langle \lambda(y_i, \xi_i), T_p(\theta) \rangle - \Psi_\ell(T_p(\theta))). \tag{21}$$

Dropping $\theta$-independent factors and collecting exponents:

$$\propto h(\theta) \exp\left( \Psi_\ell(T_p(\theta)) + \sum_{i=1}^{n} \langle \lambda(y_i, \xi_i), T_p(\theta) \rangle - n\, \Psi_\ell(T_p(\theta)) \right), \tag{22}$$

Collecting terms and bringing the sum inside the inner product we get the $n$th step posterior:

$$p(\theta \mid y_{1:n}, \xi_{1:n}) \propto h(\theta) \exp\left( \left\langle \sum_{i=1}^{n} \lambda(y_i, \xi_i),\ T_p(\theta) \right\rangle - (n-1)\, \Psi_\ell(T_p(\theta)) \right). \tag{23}$$

## C. Proof of Element-Wise Monotonicity in the Exposure Function $c(\xi)$

**Claim:** Assume a likelihood of the form

$$p(y \mid \theta, c(\xi)) = h(y) \exp\{\langle S(y), \eta(\theta)\rangle - \langle c(\xi),\, A(\eta(\theta))\rangle\}, \tag{24}$$

where $c(\xi) \in \mathbb{R}^m_{\geq 0}$ is a continuous vector-valued exposure function and $A(\eta) = (A_1(\eta), \ldots, A_m(\eta)) \in \mathbb{R}^m$. Then EIG$(\xi)$ (with respect to $\theta$) is monotone in $c(\xi)$ with respect to the component-wise order: for $c, c' \in \mathbb{R}^m_{\geq 0}$,

$$c' \succeq c \iff c'_j \geq c_j \text{ for all } j = 1, \ldots, m.$$

*Proof.* We now work in terms of the sufficient statistic, rather than $y$ directly. Note we no longer explicitly express the dependence of the exposure function, $c(\xi)$, on $\xi$. Let us denote the random variable with density proportional to $p(S_c \mid \theta, c) = h_S(S_c, \xi) \exp\{\langle \eta(\theta), S_c\rangle - \langle c, A(\eta(\theta))\rangle\}$, by $S_c$, where $h_S$ is the base measure of the sufficient statistic, i.e. the pushforward of $h$ under $T$. Defining the function

$$Z_c(\eta) = \int h_S(t, \xi) e^{\langle \eta, t\rangle}\, dt = \int h(y, \xi) e^{\langle S(y), \eta\rangle}\, dy = \exp\left(\langle c, A(\eta)\rangle\right).$$

we can express the characteristic function of $S_c$ as

$$\varphi_{S_c}(t) := \mathbb{E}\left[e^{i\langle t, S_c\rangle}\right] = \frac{Z_c(\eta + it)}{Z_c(\eta)} = \exp\left(\langle c,\, A(\eta + it) - A(\eta)\rangle\right).$$

Set $\psi(t) := A(\eta + it) - A(\eta)$; then $\varphi_{S_c}(t) = \exp(\langle c, \psi(t)\rangle)$. Therefore, for any $c_1, c_2 \in \mathbb{R}^m_{\geq 0}$,

$$\varphi_{S_{c_1+c_2}}(t) = \exp\left(\langle c_1 + c_2, \psi(t)\rangle\right) \tag{25}$$
$$= \exp\left(\langle c_1, \psi(t)\rangle\right) \exp\left(\langle c_2, \psi(t)\rangle\right) \tag{26}$$
$$= \varphi_{S_{c_1}}(t)\, \varphi_{S_{c_2}}(t) \tag{27}$$
$$= \varphi_{S_{c_1} * S_{c_2}}(t), \tag{28}$$

where the last equality uses that the characteristic function of a convolution is the product of characteristic functions. By uniqueness of characteristic functions we conclude that,

$$S_{c_1+c_2} \overset{d}{=} S_{c_1} + S'_{c_2},$$

where $S'_{c_2}$ is independent of $S_{c_1}$ conditional on $\theta$ and has the same distribution as $S_{c_2}$. We say the family $\{S_c\}_{c \in \mathbb{R}^m_{\geq 0}}$ forms a convolution semigroup. By this convolution semigroup property, we can conclude an independent increments property, namely, for $c, c' \in \mathbb{R}^m_{\geq 0}$ with $c' \succeq c$, we may conclude that,

$$S_{c'} \overset{d}{=} S_c + \Delta_{c'-c},$$

where $\Delta_{c'-c}$ is independent of $S_c$ conditional on $\theta$ (Bertoin, 1996). Assuming the prior on $\theta$ is independent of $c$, we can use this independent-increments representation, to apply the chain rule for mutual information,

$$I(\theta; S_{c'}) = I(\theta; S_c, \Delta_{c'-c}) = I(\theta; S_c) + I(\theta; \Delta_{c'-c} \mid S_c).$$

Where $I(\theta; S_c) := \text{EIG}(c)$. Since conditional mutual information is nonnegative,

$$\text{EIG}(c') - \text{EIG}(c) = I(\theta; \Delta_{c'-c} \mid S_c) \geq 0.$$

Thus $\text{EIG}(c') \geq \text{EIG}(c)$ for any $c' \succeq c$. $\qquad\square$

## D. Bregman Legendre Transform Representation of KL-Divergence for EF

The following proposition and proof have been adapted from (Nielsen, 2023),

**Theorem D.1** (Dual Bregman representations of KL). *Let $\{p_\lambda(\theta)\}_{\lambda \in \Lambda}$ be a regular exponential family in canonical form*

$$p_\lambda(\theta) = h(\theta) \exp\big(\langle \lambda, \theta \rangle - \Psi(\lambda)\big), \tag{29}$$

*where $\Psi : \Lambda \to \mathbb{R}$ is the (Legendre-type) log-partition function, $\Lambda$ is open and convex, and $\Psi$ is strictly convex and differentiable on $\Lambda$. For $\lambda_i \in \Lambda$ define the expectation (mean) parameters $\mu_i = \nabla\Psi(\lambda_i)$ and let $\Psi^*$ denote the convex conjugate of $\Psi$. Then for any $\lambda_0, \lambda \in \Lambda$*

$$\mathrm{KL}\big(p_\lambda \big\| p_{\lambda_0}\big) = D_\Psi(\lambda_0 \| \lambda) = D_{\Psi^*}(\mu \| \mu_0),$$

*where for a convex differentiable generator $f$ the Bregman divergence is*

$$D_f(u\|v) := f(u) - f(v) - \langle \nabla f(v),\, u - v \rangle.$$

*Proof.* By definition,

$$\begin{aligned}
\mathrm{KL}(p_\lambda \| p_{\lambda_0}) &= \mathbb{E}_{p_\lambda}\big[\log p_\lambda(\theta) - \log p_{\lambda_0}(\theta)\big] \\
&= \mathbb{E}_{p_\lambda}\big(\langle \lambda - \lambda_0,\, T(\theta)\rangle - \Psi(\lambda) + \Psi(\lambda_0)\big) \\
&= \langle \lambda - \lambda_0, \mathbb{E}_{p_\lambda}[T(\theta)]\rangle - \Psi(\lambda) + \Psi(\lambda_0).
\end{aligned}$$

Using $\mu = \nabla\Psi(\lambda)$,

$$\mathrm{KL}(p_\lambda \| p_{\lambda_0}) = \Psi(\lambda_0) - \Psi(\lambda) - \langle \nabla\Psi(\lambda),\, \lambda_0 - \lambda\rangle = D_\Psi(\lambda_0 \| \lambda).$$

Now, define the convex conjugate,

$$\Psi^*(\mu) = \sup_{\lambda \in \Lambda}\{\langle \mu, \lambda\rangle - \Psi(\lambda)\}.$$

Under the Legendre-type assumptions the supremum is attained uniquely at the $\lambda$ with $\mu = \nabla\Psi(\lambda)$, and the reciprocal relations hold:

$$\mu = \nabla\Psi(\lambda), \qquad \lambda = \nabla\Psi^*(\mu), \qquad \Psi^*(\mu) = \langle \lambda, \mu\rangle - \Psi(\lambda).$$

We now wish to re-express $D_\Psi(\lambda_0 \| \lambda)$ in mean coordinates. Substitute $\Psi(\lambda) = \langle \lambda, \mu\rangle - \Psi^*(\mu)$ (valid when $\mu = \nabla\Psi(\lambda)$) into the formula for the KL,

$$\begin{aligned}
\mathrm{KL}(p_\lambda \| p_{\lambda_0}) &= \big(\langle \lambda_0, \mu_0\rangle - \Psi^*(\mu_0)\big) - \big(\langle \lambda, \mu\rangle - \Psi^*(\mu)\big) - \langle \mu,\, \lambda_0 - \lambda\rangle \\
&= \Psi^*(\mu) - \Psi^*(\mu_0) - \langle \lambda_0,\, \mu - \mu_0\rangle.
\end{aligned}$$

Using $\lambda_0 = \nabla\Psi^*(\mu_0)$, the right-hand side is precisely,

$$\Psi^*(\mu) - \Psi^*(\mu_0) - \big\langle \nabla\Psi^*(\mu_0),\, \mu - \mu_0\big\rangle = D_{\Psi^*}(\mu \| \mu_0).$$

Hence,

$$\mathrm{KL}(p_\lambda \| p_{\lambda_0}) = D_\Psi(\lambda_0 \| \lambda) = D_{\Psi^*}(\mu \| \mu_0).$$

$\square$

## E. EPIG

The Expected Predictive Information Gain (EPIG) can be written as (Smith et al., 2023),

$$\mathbb{E}_{p_*(\xi_*)}\left[\mathrm{KL}\left[p_\phi\left(y, y_* \mid \xi, \xi_*\right) \big\| p_\phi(y \mid \xi)p_\phi\left(y_* \mid \xi_*\right)\right]\right]$$

from which it is clear that if the joint distribution, and the product of the marginals are in the same EF then EPIG becomes singly intractable, using the formula for the KL-divergence of two EF distributions in the same family (Nielsen & Nock, 2010). This is very restrictive, but is known to hold for Gaussian likelihoods, empowering recent applications to Gaussian Process Regression (GPR) (Takeno et al., 2025). The forms for the singly-nested and doubly-nested estimators for GPR are presented below,

### E.1. Singly-Intractable and Doubly Intractable EPIG forms

The following is taken directly from the Appendix of (Smith et al., 2023).

If we cannot integrate over $y$ and $y_*$ analytically, we can revert to nested Monte Carlo estimation. For this we first note that, using Equation 2, we can sample $y, y_* \sim p_\phi(y, y_* \mid x, x_*)$ exactly by drawing a $\theta$ and then any $y$ conditioned on this $\theta$. By also drawing samples for $\theta$, we can then construct the estimator

$$\text{EPIG}(x) \approx \frac{1}{M} \sum_{j=1}^{M} \log \frac{\frac{1}{K} \sum_{i=1}^{K} p_\phi(y^j \mid x, \theta_i)\, p_\phi(y_*^j \mid x_*^j, \theta_i)}{\left(\frac{1}{K} \sum_{i=1}^{K} p_\phi(y^j \mid x, \theta_i)\right)\left(\frac{1}{K} \sum_{i=1}^{K} p_\phi(y_*^j \mid x_*^j, \theta_i)\right)}, \tag{30}$$

where $x_*^j \sim p_*(x_*)$, $y^j, y_*^j \sim p_\phi(y, y_* \mid x, x_*^j)$ and $\theta_i \sim p_\phi(\theta)$. Subject to some weak assumptions on $p_\phi$, this estimator converges as $K, M \to \infty$.

### E.2. Gaussian predictive distribution

In the batch case of GPR, we obtain a singly-intractable form for EPIG. Consider a joint predictive distribution that is multivariate Gaussian with mean vector $\boldsymbol{\mu}$ and block covariance matrix $\boldsymbol{\Sigma}$:

$$p_\phi(\mathbf{y}, \mathbf{y}_* \mid \mathbf{x}, \mathbf{x}_*) = \mathcal{N}(\boldsymbol{\mu}, \boldsymbol{\Sigma}) = \mathcal{N}\left(\boldsymbol{\mu}, \begin{bmatrix} K_{\mathbf{xx}} & K_{\mathbf{xx}_*} \\ K_{\mathbf{x}_*\mathbf{x}} & K_{\mathbf{x}_*\mathbf{x}_*} \end{bmatrix}\right),$$

where $K_{\mathbf{xx}} \in \mathbb{R}^{5 \times 5}$ has entries $[K_{\mathbf{xx}}]_{ij} = \text{cov}(x_i, x_j)$, and likewise for $K_{\mathbf{xx}_*}$ and $K_{\mathbf{x}_*\mathbf{x}_*}$, so $\boldsymbol{\Sigma} \in \mathbb{R}^{10 \times 10}$.

The mutual information between $\mathbf{y}$ and $\mathbf{y}_*$ given $\mathbf{x}$ and $\mathbf{x}_*$ is a closed-form function of $\boldsymbol{\Sigma}$. Using the differential entropy of a multivariate Gaussian, $\text{H}[\mathcal{N}(\cdot, A)] = \frac{1}{2}\log\det(2\pi e\, A)$:

$$\begin{aligned} I(\mathbf{y}; \mathbf{y}_* \mid \mathbf{x}, \mathbf{x}_*) &= \text{H}\big[p_\phi(\mathbf{y} \mid \mathbf{x})\big] + \text{H}\big[p_\phi(\mathbf{y}_* \mid \mathbf{x}_*)\big] - \text{H}\big[p_\phi(\mathbf{y}, \mathbf{y}_* \mid \mathbf{x}, \mathbf{x}_*)\big] \\ &= \tfrac{1}{2}\log\det(2\pi e\, K_{\mathbf{xx}}) + \tfrac{1}{2}\log\det(2\pi e\, K_{\mathbf{x}_*\mathbf{x}_*}) - \tfrac{1}{2}\log\det(2\pi e\, \boldsymbol{\Sigma}) \\ &= \tfrac{1}{2}\log \frac{\det K_{\mathbf{xx}}\ \det K_{\mathbf{x}_*\mathbf{x}_*}}{\det \boldsymbol{\Sigma}}. \end{aligned}$$

Applying the block-matrix determinant identity to $\boldsymbol{\Sigma}$,

$$\det \boldsymbol{\Sigma} = \det K_{\mathbf{xx}} \cdot \det \underbrace{\big(K_{\mathbf{x}_*\mathbf{x}_*} - K_{\mathbf{x}_*\mathbf{x}}\, K_{\mathbf{xx}}^{-1}\, K_{\mathbf{xx}_*}\big)}_{S_{\mathbf{x}_* \mid \mathbf{x}}},$$

where $S_{\mathbf{x}_* \mid \mathbf{x}}$ is the $5 \times 5$ Schur complement (the posterior covariance of $\mathbf{y}_*$ given $\mathbf{y}$), we obtain:

$$\begin{aligned} I(\mathbf{y}; \mathbf{y}_* \mid \mathbf{x}, \mathbf{x}_*) &= \tfrac{1}{2}\log \frac{\det K_{\mathbf{xx}}\ \det K_{\mathbf{x}_*\mathbf{x}_*}}{\det K_{\mathbf{xx}} \cdot \det S_{\mathbf{x}_* \mid \mathbf{x}}} \\ &= \tfrac{1}{2}\log \frac{\det K_{\mathbf{x}_*\mathbf{x}_*}}{\det S_{\mathbf{x}_* \mid \mathbf{x}}} \\ &= \tfrac{1}{2}\log \frac{\det K_{\mathbf{x}_*\mathbf{x}_*}}{\det\big(K_{\mathbf{x}_*\mathbf{x}_*} - K_{\mathbf{x}_*\mathbf{x}}\, K_{\mathbf{xx}}^{-1}\, K_{\mathbf{xx}_*}\big)}. \end{aligned}$$

We can then estimate EPIG using samples $\mathbf{x}_*^j \sim p_*(\mathbf{x}_*)$, where each $\mathbf{x}_*^j = (x_{*1}^j, \ldots, x_{*5}^j)^\top$ is a batch of five test points:

$$\text{EPIG}(\mathbf{x}) = \mathbb{E}_{p_*(\mathbf{x}_*)}[I(\mathbf{y}; \mathbf{y}_* \mid \mathbf{x}, \mathbf{x}_*)]$$

$$\approx \frac{1}{M} \sum_{j=1}^{M} I(\mathbf{y}; \mathbf{y}_* \mid \mathbf{x}, \mathbf{x}_*^j)$$

$$= \frac{1}{2M} \sum_{j=1}^{M} \log \frac{\det K_{\mathbf{x}_*^j \mathbf{x}_*^j}}{\det\big(K_{\mathbf{x}_*^j \mathbf{x}_*^j} - K_{\mathbf{x}_*^j \mathbf{x}} K_{\mathbf{x}\mathbf{x}}^{-1} K_{\mathbf{x}\mathbf{x}_*^j}\big)}. \tag{31}$$

## F. Exponential Family Forms of Models in Section 7

### Poisson Likelihood and Gamma Prior

A Poisson likelihood for an observation $y \in \{0, 1, 2, \dots\}$, with rate parameter $\theta > 0$ is given by,

$$p(y \mid \theta) = \frac{\theta^y e^{-\theta}}{y!}. \tag{32}$$

Define the canonical (natural) parameter $\eta = \log \theta \iff \theta = e^\eta$. Then we can write the likelihood in exponential-family form,

$$p(y \mid \eta) = \exp\{\eta y - e^\eta - \log(y!)\} \propto \exp\{\eta y - e^\eta\} \tag{33}$$

Thus the Poisson distribution belongs to a one-parameter exponential family with sufficient statistic, $S(y) = y$, and log-partition $A(\eta) = e^\eta$. Now, if the rate parameter is instead given by $c(\xi)\theta$, then we can take the same sufficient statistic, $S(y) = y$, but now with log-partition $A(\eta) = c(\xi)e^\eta$.

For the prior we take the canonical Diaconis–Ylvisaker (DY) prior on the natural parameter $\eta$, with density

$$\pi(\eta \mid \alpha, \beta) \propto \exp\{\alpha\eta - \beta A(\eta)\}, \qquad \alpha > 0, \ \beta > 0. \tag{34}$$

Now letting $A(\eta) = e^\eta$, we obtain, up to constants in $\eta$,

$$\pi(\eta \mid \alpha, \beta) \propto \exp\{\alpha\eta - \beta e^\eta\}. \tag{35}$$

We can compute the log-partition of the DY prior as follows. Let,

$$\Psi(\alpha, \beta) := \log \int_{-\infty}^{\infty} \exp\{\alpha\eta - \beta e^\eta\} \, d\eta. \tag{36}$$

using the change of variables $\theta = e^\eta$, so that $\eta = \log\theta$ and $d\eta = d\theta/\theta$, we have

$$\int_{-\infty}^{\infty} \exp\{\alpha\eta - \beta e^\eta\} \, d\eta = \int_{0}^{\infty} \exp\{\alpha\log\theta - \beta\theta\} \frac{d\theta}{\theta} \tag{37}$$

$$= \int_{0}^{\infty} \theta^\alpha e^{-\beta\theta} \frac{d\theta}{\theta} \tag{38}$$

$$= \int_{0}^{\infty} \theta^{\alpha-1} e^{-\beta\theta} \, d\theta \tag{39}$$

$$= \beta^{-\alpha} \Gamma(\alpha). \tag{40}$$

Therefore,

$$\Psi(\alpha, \beta) = \log\Gamma(\alpha) - \alpha\log\beta. \tag{41}$$

The *mean parametrisation* of the DY prior is defined as

$$m(\alpha, \beta) := \mathbb{E}_{\pi(\eta|\alpha,\beta)}[T(\eta)],$$

where the sufficient statistic is

$$T(\eta) = \begin{bmatrix} \eta \\ e^\eta \end{bmatrix}.$$

Consequently, the mean parameter takes the form

$$m(\alpha, \beta) = \begin{bmatrix} \mathbb{E}[\eta] \\ \mathbb{E}[e^\eta] \end{bmatrix}.$$

Let $\theta = e^\eta$. As shown below, $\theta \sim \mathrm{Gamma}(\alpha, \beta)$, from which it follows that

$$\mathbb{E}[e^\eta] = \mathbb{E}[\theta] = \frac{\alpha}{\beta}, \qquad \mathbb{E}[\eta] = \mathbb{E}[\log \theta] = \psi(\alpha) - \log \beta,$$

where $\psi(\cdot)$ denotes the digamma function. Therefore, the mean-parameter vector is

$$m(\alpha, \beta) = \begin{bmatrix} \psi(\alpha) - \log \beta \\ \alpha/\beta \end{bmatrix}.$$

It is well known that the Gamma distribution is a conjugate prior for the rate parameter of a Poisson likelihood. This can be seen directly via a change of variables with $\theta = e^\eta$. The induced prior on $\theta$ is

$$\pi(\theta \mid \alpha, \beta) = \pi(\eta) \left| \frac{d\eta}{d\theta} \right| \propto \exp\{\alpha \log \theta - \beta\theta\} \frac{1}{\theta} = \theta^{\alpha-1} e^{-\beta\theta},$$

which is the kernel of a Gamma distribution with shape parameter $\alpha$ and rate parameter $\beta$. Hence,

$$\theta \mid \alpha, \beta \sim \mathrm{Gamma}(\alpha, \beta).$$

In practice, we would typically invoke this well-known conjugacy and directly use the corresponding log-partition function and mean parametrisation, rather than deriving them explicitly as done here for exposition.

**Wishart Likelihood and Inverse–Wishart Prior**

Let $Y \in \mathbb{R}^{p \times p}$ be a symmetric positive definite observation distributed according to a Wishart law with $n(\xi)$ degrees of freedom and (unknown) scale matrix $\Sigma \succ 0$. Using the common scale parametrisation, the Wishart density is

$$p(Y \mid \Sigma, n) = \frac{|Y|^{\frac{n-p-1}{2}} \exp\{-\frac{1}{2} \mathrm{tr}(\Sigma^{-1}Y)\}}{2^{\frac{np}{2}} |\Sigma|^{\frac{n}{2}} \Gamma_p(\frac{n}{2})},$$

where $\Gamma_p(\cdot)$ is the multivariate Gamma function and $n \geq p$ (we write $n := n(\xi)$ when the d.o.f. depends on $\xi$). The expectation of $Y$ under this parameterisation is $\mathbb{E}[Y] = n\Sigma$.

To display this in exponential–family (canonical) form, take the sufficient statistic

$$S(y) = \begin{bmatrix} \log|Y| \\ Y \end{bmatrix},$$

and the canonical (natural) parameter

$$\eta = \begin{bmatrix} \eta_1 \\ \eta_2 \end{bmatrix} = \begin{bmatrix} \frac{n-p-1}{2} \\ -\frac{1}{2}\Sigma^{-1} \end{bmatrix}.$$

With these choices the Wishart density can be written as, up to constants in $\eta$

$$p(Y \mid \eta) \propto \exp\{\langle S(Y), \eta \rangle - A(\eta)\},$$

where the inner product is

$$\langle S(Y), \eta \rangle = \eta_1 \log|Y| + \mathrm{tr}(\eta_2 Y)$$

with $\Sigma$ recovered from $\eta_2$ via $\Sigma = -\frac{1}{2}\eta_2^{-1}$. Equivalently, expressing $A$ directly in terms of $\eta$,

$$A(\eta) = \frac{np}{2}\log 2 - \frac{n}{2}\log\big|-2\eta_2\big| + \log\Gamma_p\Big(\frac{n}{2}\Big) = \frac{np}{2}\log 2 + \frac{n}{2}\log|\Sigma| + \log\Gamma_p\Big(\frac{n}{2}\Big),$$

where $n = 2\eta_1 + p + 1$ as implied by the mapping between $\eta_1$ and $n$.

Thus the Wishart is a (multi-parameter) exponential family with sufficient statistics $\log|Y|$ and $Y$, natural parameter $\big((n-p-1)/2,\ -\frac{1}{2}\Sigma^{-1}\big)$, and log-partition function $A(\eta)$ as above.

**Inverse–Wishart prior.** We let $p = 10$, and place an inverse–Wishart prior on the unknown scale matrix $\Sigma$,

$$\pi(\Sigma \mid \Lambda, \nu) \propto |\Sigma|^{-\frac{\nu+p+1}{2}} \exp\Big\{-\tfrac{1}{2}\operatorname{tr}(\Lambda\,\Sigma^{-1})\Big\}, \qquad \Lambda \succ 0,\ \nu > 0,$$

where $\Lambda$ is the scale (inverse-scale) hyperparameter and $\nu > 0$ denotes the degrees of freedom. The corresponding log-partition (log-normalising) function of the prior as

$$\Psi(\Lambda, \nu) := \log\Big(2^{\frac{\nu p}{2}} |\Lambda|^{-\frac{\nu}{2}} \Gamma_p\Big(\frac{\nu}{2}\Big)\Big).$$

Equivalently, the prior can be written in exponential–family form (as a distribution over $\Sigma$) using sufficient statistics

$$T_\pi(\Sigma) = \begin{bmatrix} \log|\Sigma| \\ \Sigma^{-1} \end{bmatrix}, \qquad \text{and natural parameter} \qquad \eta_\pi = \begin{bmatrix} -\frac{\nu+p+1}{2} \\ -\frac{1}{2}\Lambda \end{bmatrix},$$

so that

$$\pi(\Sigma \mid \Lambda, \nu) = \exp\big\{\langle \eta_\pi, T_\pi(\Sigma)\rangle - \Psi(\Lambda, \nu)\big\}.$$

**Conjugacy and posterior.** Let $Y \sim \operatorname{Wishart}(n, \Sigma)$ denote a single observation from the Wishart likelihood with $n = n(\xi)$ degrees of freedom. Up to terms not depending on $\Sigma$, the likelihood can be written as

$$p(Y \mid \Sigma) \propto |\Sigma|^{-\frac{n}{2}} \exp\Big\{-\tfrac{1}{2}\operatorname{tr}(Y\,\Sigma^{-1})\Big\}.$$

Multiplying by the inverse–Wishart prior yields a posterior density proportional to

$$|\Sigma|^{-\frac{\nu+p+1}{2}} |\Sigma|^{-\frac{n}{2}} \exp\Big\{-\tfrac{1}{2}\operatorname{tr}\big((\Lambda + Y)\Sigma^{-1}\big)\Big\},$$

which can be recognised as the kernel of an inverse–Wishart distribution. Hence,

$$\Sigma \mid Y \sim \operatorname{InvWishart}\big(\Lambda + Y,\ \nu + n\big).$$

**Mean parametrisation of the prior.** The mean parametrisation is defined via the expectations of the prior sufficient statistics,

$$m(\Lambda, \nu) := \mathbb{E}_{\pi(\Sigma \mid \Lambda, \nu)}\big[T_\pi(\Sigma)\big] = \begin{bmatrix} \mathbb{E}[\log|\Sigma|] \\ \mathbb{E}[\Sigma^{-1}] \end{bmatrix}.$$

Using standard properties of the inverse–Wishart distribution (equivalently, that $\Sigma^{-1} \sim \operatorname{Wishart}(\Lambda^{-1}, \nu)$), we obtain

$$\mathbb{E}[\Sigma^{-1}] = \nu\,\Lambda^{-1},$$

and

$$\mathbb{E}[\log|\Sigma|] = \log|\Lambda| - \sum_{j=1}^{p} \psi\Big(\frac{\nu+1-j}{2}\Big) - p\log 2,$$

where $\psi(\cdot)$ denotes the digamma function and the expectation of $\log|\Sigma|$ is finite for $\nu > p - 1$. Consequently, the mean–parameter vector of the inverse–Wishart prior is

$$m(\Lambda, \nu) = \begin{bmatrix} \log|\Lambda| - \sum_{j=1}^{p}\psi\big(\frac{\nu+1-j}{2}\big) - p\log 2 \\ \nu\,\Lambda^{-1} \end{bmatrix}.$$

# G. Experimental Details

### G.1. Gaussian Likelihood and Prior for Convergence Plots

We place a Gaussian prior directly on the parameter $\theta$,

$$\theta \sim \mathcal{N}(\mathbf{0},\ \mathbf{I}_d), \tag{42}$$

with zero mean and isotropic covariance $\mathbf{I}_d$.

**Likelihood**    Given the parameter $\theta$ and a design point $\xi$, the observation $y$ follows a Gaussian likelihood,

$$y \mid \theta, \xi \sim \mathcal{N}(\xi\theta,\ \mathbf{I}_d), \tag{43}$$

where $\mathbf{I}_d$ is the identity noise covariance.

**Design Variables**    The experimental design variable is a scalar $\xi \in \mathbb{R}$, fixed at $\xi = 1.5$

Under the Gaussian likelihood and prior, the EIG admits a closed-form expression,

$$\mathrm{EIG}(\xi) = \frac{1}{2} \log \det\left(\mathbf{I}_d + \xi\xi^\top \Sigma_0\right), \tag{44}$$

where $\Sigma_0$ is the prior covariance.

### G.2. Poisson Likelihood and Gamma Prior

We model the natural parameter $\eta = \log\theta$ and assume a log-Gamma prior, corresponding to a Gamma prior on $\theta \sim$ Gamma$(a_0, b_0)$, with shape and rate parameters $a_0 = 20,\ b_0 = 10$.

**Design Variables**    The experimental design variable, $\xi = \{\xi_1, \xi_2, \xi_3, \xi_4\} \in \mathbb{R}^8$, is a collection of 4 2-D co-ordinates, defining a rhombus. The likelihood given $\theta$ and $\xi$ is Poisson with rate $\theta \times \mathrm{Area}(\xi)$, where $\mathrm{Area}(\xi)$ is the area of the rhombus defined by $\xi$. We take our utility function to be $\mathrm{U}(\xi) = \mathrm{EIG}(\xi) - \gamma\, \mathrm{C}(\xi)$, where $C(\xi) = \sum_i ||\xi_i||_2$, and $\gamma = 0.05$.

We treat $\xi$ as a continuous variable and optimize it using gradients. Specifically, we take the respective **NMC**, **PCE**, and **SI** estimator and taking gradients with respect to $\xi$. Before doing this we apply a softening of the Poisson sampling procedure, as detailed in (Shchur et al., 2020), with temperature 0.5, which we found to have low variance and bias.

### G.3. Wishart Likelihood and Inverse Wishart Prior

We model the scale matrix $\Sigma \in \mathbb{S}_+^p$ of the Wishart likelihood and place an inverse-Wishart prior on $\Sigma$. Specifically, the observation model is

$$Y \mid \Sigma, \xi \sim \mathrm{Wishart}(n(\xi),\ \Sigma),$$

where $n(\xi)$ denotes the degrees of freedom and $\Sigma$ is the unknown positive-definite scale matrix. We assume the prior

$$\Sigma \sim \mathrm{Inv\text{-}Wishart}(a_0, B_0),$$

with degrees of freedom $a_0 = 2p + 5$, and scale matrix given by $B_0 = I_p$. We take $p = 10$ in our experiments, meaning $\Sigma$ and $Y$ are $10 \times 10$ matrices.

**Design Variables**    We optimize for $n(\xi) = \xi$ by calculating $U(\xi) = \mathrm{EIG}(\xi)$-$\gamma\xi + 1$, where $\gamma = 0.1$ over the grid $\xi \in \{p, p + 10, \ldots, 20p\}$. We take the $\mathrm{argmax}$ of $U(\xi)$. The addition of 1 is to ensure the utility is not consistently negative, which leads to an unaesthetic plot, but of course, does not affect the optimization.

### G.4. GPR

We place a zero-mean Gaussian process prior on the latent function $f$,

$$f(\cdot) \sim \mathcal{GP}(0, k(\cdot, \cdot)),$$

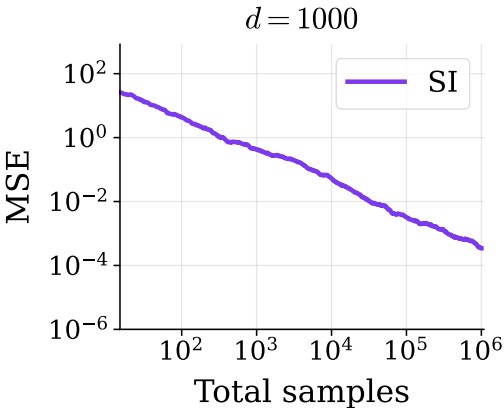

*Figure 6.* SI estimator empirical convergence rate in 1000D for the same linear Gaussian regression experiment set up as in Section 7

where the covariance function $k$ is chosen to be a squared exponential (RBF) kernel,

$$k(x, z) = \sigma^2 \exp\left(-\frac{1}{2\ell^2}\|x - z\|^2\right).$$

with fixed hyperparameters $\ell = 0.5$, $\sigma^2 = 1.0$.

**Design Variables**   The experimental design is the collection of 5 2-D input locations, $X_\xi = \{x_1, \ldots, x_5\}$, which are treated as continuous optimization variables. Each design point is constrained to lie within the compact domain $[-1, 1]$, enforced by projection after every optimization step.

We initialize the batch using 5 independent draws from a uniform distribution over $[-1, 1]^2$. The design locations $X$ are optimized by gradient-based maximization of EPIG. We simply take the gradient of the MC EPIG estimator (for both the **NMC** and **SI** estimators) without the need for any reparametrization trick. We use Adam with learning rate $\eta = 0.05$, and run the optimization for 100 gradient steps. At each iteration, the design points are updated and subsequently clamped to the feasible domain.

## H. Singly-Intractable Estimator Scales to 1000D

See Figure 6 showing the **SI** estimator scaling to 1000D, for the same Gaussian linear regression problem as described in Section 7.

## I. Expected Divergence with a Variational Posterior in the EF

**Expected 'Variational KL'**   One might use Theorem 3.1 to see that for a fixed $q(\theta)$ (not necessarily our true prior, or the current variational distribution), up to constants in $\xi$,

$$\text{EIG}(\xi) = \text{E}_{p(y)} \text{KL}\left[p(\theta|y, \xi)\|q(\theta)\right] \tag{45}$$
$$\approx \text{E}_{p(y)} \text{KL}\left[q(\theta|y, \xi)\|q(\theta)\right] := \text{EVKL}(\xi), \tag{46}$$

for $q(\theta|y, \xi)$ some variational distribution 'close' to the true posterior $p(\theta|y, \xi)$. Note this is a slight generalization of the objective functions discussed in (Bickford Smith et al., 2025; Khan, 2025), as $q(\theta)$ is not assumed to represent our current belief distribution). Under mild regularity conditions, as the variational posterior approaches the true posterior, $\text{EVKL}(\xi) - \text{EIG}(\xi)$ approaches zero.

By taking $q(\theta)$ to be in the same EF as the variational posterior approximation, $q(\theta|y, \xi)$, $\text{EVKL}(\xi)$ is singly-intractable, *assuming* that the parameters of the variational posterior are known. Unfortunately, variational updates for a given datum are unlikely to be given in a closed-form update, and are usually the result of a (stochastic) optimization procedure e.g, the Bayesian Learning Rule (Khan & Rue, 2023)). Hence, the use of $\mathbb{E}_{p(y)}[\text{KL}(q(\theta|y, \xi) \| q(\theta))]$ over the EIG only offers

computational benefits when using very cheap variational updates, e.g, variants of BONG (Jones et al., 2024), limiting its practicality as an acquisition function.

