# OpenReview forum: "Tractable Expected Information Gains for Exponential Family Posteriors"
_ICML.cc/2026/Conference — ICML 2026 regular_

### Official Review · Reviewer_UYRD · 2026-02-20

**Soundness:** 3
**Presentation:** 3
**Significance:** 3
**Originality:** 3
**Overall Recommendation:** 4
**Confidence:** 2

**Summary:**

This paper addresses the "double intractability" issue in Bayesian Experimental Design (BED), where estimating Expected Information Gain (EIG) typically requires biased and computationally expensive Nested Monte Carlo (NMC) procedures. The key theoretical contribution is a proof that when the posterior belongs to the Exponential Family (EF) and the experimental design affects the posterior only through its natural parameters, the EIG simplifies from a doubly intractable expectation to a singly intractable form.

**Compliance With Llm Reviewing Policy:**

Affirmed.

**Key Questions For Authors:**

1. If we use Variational Inference (VI) to approximate the posterior, how much does the SI estimator suffer if the approximation is a bit off? Is there a risk that the optimization will just drift toward areas where the variational model is poorly calibrated?

2. How’s the gradient variance when the design space gets crowded? Do we need to throw in extra variance reduction tricks to keep things stable in high dimensions?

**Limitations:**

see weakness above.

**Strengths And Weaknesses:**

#### **Strengths**
* It seems solid beyond standard Gaussian assumptions. Showing that EF posteriors are enough for the math to collapse is a neat theoretical result, but I did not verify the derivations in detail.

* The SI estimator is a much more elegant solution than nested sampling—it’s faster and skips the usual bias issues.

* I also appreciate that it works for some non-conjugate setups; keeping the posterior in the EF gives you a decent amount of modeling room.

#### **Weaknesses**
* The whole method depends on the posterior being an EF and the design only touching the natural parameters. That’s a pretty specific niche, which might limit where this can actually be used in the real world.

* In simpler cases, EIG might just become monotonic or convex. If the "optimal" design is always just pushing against the search boundary, the whole optimization framework becomes a bit moot.

---

> ### Author Rebuttal · Authors · 2026-03-31
>
> We would like to thank you for your thoughtful review. We first address the second point raised in the weaknesses section.
>
> > “In simpler cases, EIG might just become monotonic or convex. If the ‘optimal’ design is always just pushing against the search boundary, the whole optimization framework becomes a bit moot.”
>
> In cases where an experimental design has an associated variable cost, we may instead optimize a “value of information” objective (e.g., EIG plus a cost function). In such common settings, it remains highly beneficial to use the low bias/variance singly intractable (SI) estimators we propose, even if the EIG alone is monotonic or convex.
>
> More importantly, identifying when the EIG becomes degenerate has previously been a major open question. Our work provides a clear mathematical characterization of this phenomenon, offering guidance on how to avoid degeneracy and shedding light on many classical problems. We therefore respectfully disagree that this is a weakness, and instead view it as a key contribution (as also noted by other reviewers).
>
> ---
>
> We now address your questions.
>
> > “If we use Variational Inference (VI) to approximate the posterior, how much does the SI estimator suffer if the approximation is a bit off? Is there a risk that the optimization will drift toward poorly calibrated regions?”
>
> Hypothetically, one could use EF variational approximations to the posterior and maximize the expected KL divergence between the current and next-step posterior. Please see the paragraph starting on line 296 (“One might ask if there are useful alternatives to variational bounds when we only assume an EF variational posterior”) and Appendix F. However, this approach has several shortcomings, and we do not see an advantage over BA bounds (Foster et al., 2019).
>
> Although a full evaluation is beyond the scope of this work, preliminary experiments in an active learning setting suggest that optimizing the “expected variational KL” may indeed exploit poorly calibrated regions. This connects to recent work on optimization under misspecified likelihoods (Saravanan et al., 2026).
>
> However, there are some interesting alternative ways our results could potentially be exploited when the likelihood is not exactly of required form, such as using control variates or a basis expansion of the likelihood.  Please see our response to Reviewer Zudr for more information.
>
> > “How’s the gradient variance when the design space gets crowded? Do we need additional variance reduction tricks in high dimensions?”
>
> Please may we seek clarification on what is meant by “the design space gets crowded”?
>
> Additional variance reduction tricks should generally not be needed in high dimensions, though, as we find that our estimators offer particularly significant gains in such settings, with traditional nested estimators breaking down (Rainforth et al., 2018). To empirically validate this, we have created new convergence plots for a problem of varying dimensionality. Specifically, we take a Gaussian likelihood with mean $\operatorname{diag}(\xi)\theta \in \mathbb{R}^D$ and known covariance matrix. This choice allows us to compute the ground-truth EIG analytically.
>
> Please follow this anonymous link for the resulting figures: [Convergence_Plots_PDF](https://drive.google.com/file/d/1Tz5fLxHGmnpGolJ6PwViGN_De2yL6eLf/view?usp=sharing). The plots correspond (in order) to $(D = 1, M = 100)$ (M being the number of inner NMC samples), $(D = 5, M = 10{,}000)$, $(D = 10, M = 100{,}000)$, and finally $D = 100$, with $100$, $100$, $100$, and $5$ seeds, respectively. We clearly see that $D = 10$ is already sufficiently large for NMC/PCE to break down and produce highly biased estimates, whereas our singly intractable (SI) estimator continues to converge at a rate of $O(1/T)$, where $T$ is the total number of samples.
>
> Note that we critically see only modest degradation of the SI performance with the increasing dimensionality across the plots.
>
> ---
>
> ### References
> - Foster et al. *Variational Bayesian Optimal Experimental Design*. NeurIPS 2019
> - Saravanan et al. *DIFFBED: Scaling Bayesian Experimental Design to High Dimensions*. ICLR 2026

---

### Official Review · Reviewer_o6nv · 2026-03-12

**Soundness:** 3
**Presentation:** 3
**Significance:** 3
**Originality:** 3
**Overall Recommendation:** 4
**Confidence:** 5

**Summary:**

This paper studies the computation of Expected Information Gain (EIG) in Bayesian Experimental Design. Evaluating EIG is typically difficult because it involves a nested expectation, which leads to high computational cost in standard Monte Carlo estimators. The authors show that when the posterior belongs to an exponential family and the experimental design affects only the natural parameters, both the EIG and its gradient can be written as singly-nested expectations. This leads to unbiased estimators that avoid the usual nested Monte Carlo structure. The paper also discusses certain degeneracies that may arise when optimizing EIG under these models. The proposed estimators are evaluated on several experimental design problems, including Poisson exposure design, Wishart covariance estimation, and active Gaussian process regression, where they show improved convergence compared to standard nested Monte Carlo methods.

**Compliance With Llm Reviewing Policy:**

Affirmed.

**Final Justification:**

Regarding the authors' query on related work in conjugate models, I would like to highlight that I had mentioned, "...the derivations rely on standard exponential-family identities...". I believe they can be explicitly derived, without even refering to any literature.

Having said that, my major concerns have been addressed, and I maintain my original recommendation of weak accept.

**Key Questions For Authors:**

I have the following questions for the authors.

1. How often does the exponential-family condition assumed in the paper hold in real Bayesian experimental design problems? Clarifying this would help assess the practical scope of the proposed estimator.
2. How does the method behave in higher-dimensional design spaces or more complex models? Evidence in such settings would strengthen the case for scalability.
3. How frequently do the degeneracies identified in the analysis arise in real design problems? Understanding this would clarify the practical impact of the theoretical results.

**Limitations:**

Yes

**Strengths And Weaknesses:**

The paper looks at a well known computational challenge in Bayesian experimental design, namely the difficulty of computing Expected Information Gain (EIG). The main idea is to identify cases where the usual doubly nested expectation reduces to a singly nested one. I found this observation simple and quite neat, since it gives a clear condition under which EIG becomes much easier to estimate. The estimators derived under exponential-family posteriors are easy to interpret and extend earlier tractable cases known for Gaussian models. I also liked the discussion of degeneracy phenomena when optimizing EIG. The monotonicity and convexity results help explain when certain design problems become trivial. On the empirical side, the paper evaluates the approach on several design tasks and shows consistent improvements over nested Monte Carlo estimators. The convergence experiments make the computational gains fairly clear.

At the same time, I do have a few concerns. The main tractability result relies on the posterior belonging to an exponential family with a specific dependence on the design variable. While this includes some classical models, it is not obvious how often this structure appears in more modern Bayesian models such as deep probabilistic models or simulation-based inference settings. I also wondered about the level of novelty in the theoretical part. The arguments rely on familiar properties of exponential families and KL divergence identities, so it would help to see a clearer comparison with earlier tractable EIG results for Gaussian or conjugate models. On the empirical side, most experiments are in settings where the exponential-family assumptions hold exactly. I would be curious to see how the method behaves when these assumptions are only approximately satisfied. Finally, some parts of the derivations felt a bit dense while reading. A little more intuition around the main steps could make the paper easier to follow.

---

> ### Author Rebuttal · Authors · 2026-03-31
>
> Thank you for your detailed comments and helpful feedback. We will first address the points raised in the weaknesses section.
>
> > “The main tractability result relies on the posterior…”
>
> Please see our response to Question 1 below.
>
> > “I also wondered about the level of novelty in the theoretical part. The arguments rely on familiar... identities, so it would help to see a clearer comparison with earlier tractable EIG results for Gaussian or conjugate models.”
>
> Although our theoretical work builds upon familiar properties of the exponential family (EF), these properties have not previously been exploited to develop singly intractable estimators for the EIG or to identify when such simplifications are possible, which are core contributions of our work. In particular, we are unaware of earlier “tractable EIG results” for conjugate models beyond Gaussian models (for which the EIG is available analytically).
>
> > “I would be curious to see how the method behaves when these assumptions are only approximately satisfied.”
>
> This is an interesting question and a promising avenue for future work. In such settings, there are non-trivial choices regarding which approximation strategy one should use. Please see our response to Reviewer Zudr for discussion of three such possibilities.
>
> > “Finally, some parts of the derivations felt a bit dense while reading.”
>
> Thank you for the feedback—we will make appropriate edits to improve readability.
>
> ---
>
> We hope the following answers the questions raised.
>
> > “How often does the exponential-family condition assumed in the paper hold in real Bayesian experimental design problems?”
>
> The crucial constraint is that the likelihood must take the form of Equation 12 in Proposition 4.1. While this assumption certainly does not hold for all models, there are common examples where it does. Applications with such models include adaptive clinical trials (Atan et al., 2019), sensor placement (Krause et al., 2008), and Bayesian active learning for regression with Gaussian process models (see example in the paper), as well as stochastic last-layer BNN approaches (Snoek et al., 2015; Kristiadi et al., 2020). It will not generally hold for deep probabilistic models with additional stochastic layers or in simulation-based settings. We will add further discussion on this point to improve clarity.
>
> As noted above, our work does also though provide some interesting avenues for future investigation in cases where the assumed likelihood form does not hold exactly, as is the case in general deep probabilistic models more generally.
>
> > “How does the method behave in higher-dimensional design spaces or more complex models?”
>
> When applicable, our estimators offer significant gains in higher dimensions, where traditional NMC estimators break down (Rainforth et al., 2018). To empirically validate this, we created convergence plots for a problem of varying dimensionality. Specifically, we take a Gaussian likelihood with mean $\operatorname{diag}(\xi)\theta \in \mathbb{R}^D$ and known covariance matrix. This choice allows us to compute the ground-truth EIG analytically.
>
> Please follow this anonymous link for the resulting figures: [Convergence_Plots_PDF](https://drive.google.com/file/d/1Tz5fLxHGmnpGolJ6PwViGN_De2yL6eLf/view?usp=sharing). The plots correspond (in order) to $(D = 1, M = 100)$ (M is the inner NMC samples), $(D = 5, M = 10{,}000)$, $(D = 10, M = 100{,}000)$, and finally $D = 100$, with $100$, $100$, $100$, and $5$ seeds, respectively. We clearly see that $D = 10$ is already sufficiently large for NMC/PCE to break down and produce highly biased estimates, whereas our singly intractable (SI) estimator continues to converge at a rate of $O(1/T)$, where $T$ is the total number of samples.
>
> Note that we see only modest degradation of the SI performance with the increasing dimensionality across the plots.
>
> > “How frequently do the degeneracies identified in the analysis arise in real design problems?”
>
> Many experiments in classical experimental design exhibit some form of degeneracy. For example, in linear Gaussian models or GP regression, the EIG becomes monotonic in a known function of the design. Although it is difficult to precisely quantify how often degenerate setups arise in real-world problems, our work is the first to provide a more general mathematical characterization of when optimization becomes degenerate. This can be used to identify and subsequently refine ill-posed problems.
>
> ---
>
> ### References
>
> - Atan et al. *Sequential Patient Recruitment and Allocation for Adaptive Clinical Trials*. AISTATS 2019
> - Krause et al. *Near-Optimal Sensor Placements in Gaussian Processes: Theory, Efficient Algorithms and Empirical Studies*. JMLR 2008
> - Snoek et al. *Scalable Bayesian Optimization Using Deep Neural Networks*. ICML 2015
> - Kristiadi et al. *Being Bayesian, Even Just a Bit, Fixes Overconfidence in ReLU Networks*. ICML 2020
> - Rainforth et al. *On Nesting Monte Carlo Estimators*. PMLR 2018

---

> > ### Author Rebuttal · Reviewer_o6nv · 2026-04-03
> >
> > The rebuttal clarifies several aspects of the paper and improves the discussion of the applicability of the exponential-family assumption. The examples provided are helpful, although they also confirm that the framework does not extend to more general modern Bayesian models. The additional empirical evidence on higher-dimensional settings is useful, but would be more convincing if incorporated more systematically into the paper.
> >
> > Regarding novelty, the authors’ clarification that tractable EIG results beyond Gaussian models are limited helps contextualize the contribution. At the same time, since the derivations rely on standard exponential-family identities, a clearer positioning relative to existing work on conjugate models and related approximations would strengthen the presentation.

---

> > > ### Author Response · Authors · 2026-04-06
> > >
> > > That the class of models for which the EIG is singly-intractable does not include some ‘general modern Bayesian models’ is a true, but expected, shortcoming. Although we see our work as a complete and meaningful contribution, we are encouraged by your comments to develop our discussion of potential ways to expand the scope of applicable models, e.g., with ideas found in our reply to Reviewer Zudr.
> > >
> > > We are glad that the additional empirical evidence on higher-dimensional settings is useful. We commit to including this systematically into a revised version of the paper.
> > >
> > > Finally, could you kindly point us toward the specific papers or lines of work you feel we have omitted? We are currently unaware of relevant earlier works on conjugate models for BED, but would be eager to review any omissions and ensure our work is clearly positioned in a revised text.
> > >
> > > We will remain active should you have any further questions before the discussion period ends. If our responses have sufficiently answered your remaining questions, we kindly ask if you might consider raising your score to reflect this.

---

### Official Review · Reviewer_Zudr · 2026-03-13

**Soundness:** 3
**Presentation:** 3
**Significance:** 3
**Originality:** 3
**Overall Recommendation:** 5
**Confidence:** 3

**Summary:**

This paper considers conditions under which the expected information gain (EIG), expressible as a double integral with respect to the joint distribution of the parameter and data in a Bayesian experimental design, can be reduced to a single integral with respect to the marginal distribution of the data (conditional on experimental design). The main motivation is reduction of computation in Monte Carlo estimation of the EIG, or its gradient. Proposition 3.1 contains the main findings, which yield formulae for EIG and its gradient, when the posterior distribution belongs to an exponential family. A necessary condition is described to yield such a posterior, which is that the likelihood must belong to an exponential family, whose dependence on the design only comes through its sufficient statistic. Potential dangers of a degenerate parameterization in design are discussed.

**Compliance With Llm Reviewing Policy:**

Affirmed.

**Final Justification:**

As stated in my original review, the paper clearly expounds an important point in Bayesian experimental design. The authors' response also revealed some interesting ideas for future extensions applicable to non-conjugate models, which addressed my original comment on the one (minor) shortcoming of the work. I thus recommend the acceptance of this work to ICML.

**Key Questions For Authors:**

Are there ways to exploit your findings in the settings where the "singly intractable" estimators are not available to speed up or stabilize EIG estimation in non-conjugate models? For instance, can one devise more efficient Monte Carlo estimation in such cases by rejection/importance sampling ideas based on fitting the "nearest conjugate" models?

**Limitations:**

Yes.

**Strengths And Weaknesses:**

The strength of the paper lies in its clear exposition and structuring of the problem, sound and easy-to-understand theoretical analysis, and its discussion of possible degeneracy issues. Also, performance gains based on the "singly intractable" estimators based on a reduced integral representation, whenever available, are clear in the empirical results. The clearest limitation, which cannot be faulted on the authors, is that the current work sheds little light on the possible avenues of achieving faster and/or more stable computation in non-conjugate models to which the findings may not be applicable (though see questions below).

---

> ### Author Rebuttal · Authors · 2026-03-31
>
> Thank you for your thoughtful review!  It is great to see that our core contributions are clear, and appreciated.
>
> > “Are there ways to exploit your findings in the settings where the "singly intractable" estimators are not available to speed up or stabilize EIG estimation in non-conjugate models? For instance, can one devise more efficient Monte Carlo estimation in such cases by rejection/importance sampling ideas based on fitting the "nearest conjugate" models?”
>
> This is a very interesting question/suggestion! We see three possible ways to do this and we will include a discussion of these approaches in a revised version of the manuscript.
>
> Although it is likely not possible to collapse the EIG from a doubly-intractable to a singly-intractable quantity using importance sampling or rejection sampling, we do see a potential to increase the efficiency of the standard NMC/PCE estimator by using an EF proposal to introduce a control variate. Specifically, we can add and subtract the analytic information gain between a proposal $q(\theta)$ and some $q_0(\theta)$ in the same EF, to the true information gain. Under certain rearrangements, this is likely to reduce variance provided $q(\theta)$ is sufficiently close to the posterior. We save this investigation of this promising avenue for future work.  Thanks for the suggestion!
>
> An alternative possible approach that could also yield effective performance would be to approximate the likelihood using a basis expansion i.e, $\log p(y \mid \theta, \xi) \approx \sum_{k=1}^{K} a_k(y, \xi) \phi_k(\theta)$, after which our singly-intractable estimators become applicable. While this would inevitably add to the cost of the estimator as $K$ grows, we suspect it will often have a better compute–accuracy trade-off than PCE/NMC, particularly when the likelihood is close to being the required form. We again save detailed investigations of this exciting direction for future work.
>
> Hypothetically, one might use EF variational approximations to the posterior and target the expected KL divergence between the current and next-step posterior. Please see the paragraph starting on line 296 with “One might ask if there are useful alternatives to variational bounds when we only assume an EF variational posterior“, as well as Appendix F. However, there are several shortcomings to this, and we do not see any advantage over using BA bounds (Foster et. al, Variational Bayesian Optimal Experimental Design, NeurIPS 2019).  Thus, we think the previous two extensions are likely more promising.

---

> > ### Author Rebuttal · Reviewer_Zudr · 2026-04-02
> >
> > I thank the authors for their careful response to my questions. These ideas sound interesting enough, and I hope they may be mentioned briefly in your next iteration of the paper as future extensions. I am retaining my original score, which is to accept the paper.

---

### Official Review · Reviewer_ijZg · 2026-03-13

**Soundness:** 3
**Presentation:** 3
**Significance:** 3
**Originality:** 3
**Overall Recommendation:** 5
**Confidence:** 2

**Summary:**

This paper studies Bayesian experimental design with expected information gain (EIG) objectives and explores a class of models beyond the commonly used Gaussian setting where EIG can be computed tractably. The authors show that if the posterior distribution of the quantity of interest belongs to an exponential family and the design affects the posterior through its natural parameters, then the EIG objective admits a simplified estimator that avoids nested expectations.

**Compliance With Llm Reviewing Policy:**

Affirmed.

**Final Justification:**

The authors have adequately addressed my questions and improved my understanding.

**Key Questions For Authors:**

1. The main theorem requires the posterior over the quantity of interest to belong to an exponential family. Is this assumption common in practical experimental design problems?
2. The theorem assumes that the design affects the posterior only through the natural parameters. How restrictive this assumption is in practice?
3. Can the result apply to Bayesian optimization, e.g., entropy-based acquisition functions? Then would it require new surrogate model, different from GP?
4. Could the authors provide runtime comparisons against standard nested Monte Carlo estimators?

**Limitations:**

Yes, but more limitations about the applicability of the theoretical results in practice can be discussed.

**Strengths And Weaknesses:**

**Strengths**:
1. The paper explores exponential-family models beyond the commonly studied Gaussian setting in Bayesian experimental design. Broadening the class of models under which EIG can be computed tractably is an interesting and meaningful direction.
2. The work provides detailed theoretical analysis characterizing conditions under which EIG admits a simplified estimator, offering a deeper understanding of tractability in Bayesian experimental design.

**Weaknesses**: While the theoretical results are interesting, the practical scope of the assumptions and the computational advantages are not fully clear.

---

> ### Author Rebuttal · Authors · 2026-03-31
>
> Thank you for the thoughtful feedback and for highlighting the “interesting and meaningful direction” of the paper. We are glad that the theoretical contribution came across clearly. Below, we clarify the practical scope of our assumptions and estimators, and we will revise the paper to make these points more explicit.
>
> > “The main theorem requires the posterior over the quantity of interest to belong to an exponential family. Is this assumption common in practical experimental design problems?”
>
> The crucial constraint is that the likelihood must take the form of Equation 12 in Proposition 4.1. While this assumption certainly does not hold for all models, there are several common examples where it does. Applications with such models include adaptive clinical trials (Atan et al., 2019), sensor placement (Krause et al., 2008), and Bayesian active learning for regression with Gaussian process models (see example in the paper), as well as stochastic last-layer BNN approaches (Snoek et al., 2015; Kristiadi et al., 2020). We will add further discussion on when this assumption holds to improve clarity.
>
> An interesting direction for future work is to study what can be done when the likelihood is not of this form. One possibility is to approximate the likelihood using a basis expansion, i.e., $\log p(y \mid \theta, \xi) \approx \sum_{k=1}^{K} a_k(y, \xi) \phi_k(\theta)$, after which our singly intractable estimators become applicable. We leave a detailed investigation of this for future work.
>
> > “The theorem assumes that the design affects the posterior only through the natural parameters. How restrictive is this assumption in practice?”
>
> For a fixed EF, the only way to index members of that family is by changing the natural parameters, so this assumption is not restrictive beyond requiring that the posterior belongs to the *same* exponential family across different observations.
>
> > “Can the result apply to Bayesian optimization, e.g., entropy-based acquisition functions? Would it then require a new surrogate model, different from a GP?”
>
> This is an interesting question. Our work can, in principle, extend to BO, since BO (with information-theoretic objectives) can be viewed as a subset of BED where the parameter of interest is the location/value of the maximum. To apply our ideas, we would need the push-forward distribution of the location/value of the maximum under the surrogate model to be an EF distribution. For GPs, this push-forward distribution is not generally EF. Whether alternative surrogate models could yield EF push-forward distributions is an interesting direction for future work.
>
> > “Could the authors provide runtime comparisons against standard nested Monte Carlo estimators?”
>
> Below, we report the mean and standard deviation of runtimes for our core experiments over 5 seeds.  Note that in all cases the performance when using the SI estimator is significantly better than when using the NMC estimator for comparable sample sizes: if our primary aim is to speed up inference/optimisation, then one could reduce the number of samples used by SI instead. We will add more detailed runtime analysis to the revised paper.
>
> ---
>
> ## Wishart experiment
>
> ### SI estimator
>
> | Setting     | Runtime (s) |
> |-------------|------------|
> | outer = 100 | 1.23 ± 0.26 |
>
> ### NMC estimator
>
> | Setting                  | Runtime (s) |
> |--------------------------|------------|
> | outer = 10, inner = 10   | 1.15 ± 0.22 |
> | outer = 50, inner = 50   | 3.23 ± 0.30 |
> | outer = 250, inner = 250 | 61.00 ± 2.75 |
>
> ---
>
> ## Poisson experiment
>
> ### SI estimator
>
> | Setting     | Runtime (s) |
> |-------------|------------|
> | outer = 100 | 3.44 ± 0.04 |
>
> ### NMC estimator
>
> | Setting                  | Runtime (s) |
> |--------------------------|------------|
> | outer = 10, inner = 10   | 4.02 ± 0.06 |
> | outer = 50, inner = 50   | 11.49 ± 2.94 |
>
> ---
>
> ## EPIG optimization
>
> *Note:* In this case, we follow Bickford Smith et al. (2023) and share $\theta$ samples in the inner loop (see Appendix E). Hence, the total number of distinct GP evaluations scales like `inner + outer`.
>
> ### SI estimator
>
> | Setting      | Runtime (s) |
> |--------------|------------|
> | outer = 1000 | 0.266 ± 0.034 |
>
> ### NMC estimator
>
> | Setting                  | Runtime (s) |
> |--------------------------|------------|
> | outer = 50, inner = 20   | 0.071 ± 0.002 |
> | outer = 100, inner = 100 | 0.211 ± 0.077 |
> | outer = 500, inner = 500 | 1.094 ± 0.128 |
>
> ---
>
> ### References
>
> - Atan et al. *Sequential Patient Recruitment and Allocation for Adaptive Clinical Trials*. AISTATS 2019
> - Krause et al. *Near-Optimal Sensor Placements in Gaussian Processes: Theory, Efficient Algorithms and Empirical Studies*. JMLR 2008
> - Snoek et al. *Scalable Bayesian Optimization Using Deep Neural Networks*. ICML 2015
> - Kristiadi et al. *Being Bayesian, Even Just a Bit, Fixes Overconfidence in ReLU Networks*. ICML 2020
> - Bickford Smith et al. *Prediction-Oriented Bayesian Active Learning*. AISTATS 2023

---

> > ### Author Rebuttal · Reviewer_ijZg · 2026-04-02
> >
> > Thanks for the authors' responses. My questions are mostly addressed. It's better to add explanations on "push-forward distribution" to improve clarity.

---

### Decision · Program_Chairs · 2026-04-30

**Decision:**

Accept (regular)

**Comment:**

This paper investigates conditions under which Bayesian experimental design with the EIG objective yields updates that are only singly-, as opposed to doubly-, intractable.  The authors show that if the posterior is exponential family, and the design affects the posterior only through its natural parameters, then the EIG objective admits a simplified estimator.

Reviewers agree that this work broadens the set of models under which EIG can be computed tractably.  There is a consensus that the work is an "interesting and meaningful" (ijZg) direction.  Reviewers praise the "sound and easy-to-understand theoretical analysis" (Zudr).

As far as weaknesses, reviewers feel that the assumptions required for the theoretical analysis may be unclear.  In particular it is "not obvious how often this assumed structure appears in...modern Bayesian models" (o6nv).  Reviewers also criticize that the experimental evaluation is performed exclusively in a setting where exponential family assumptions hold exactly, and that this sheds little light on more practical non-conjugate settings.

Overall, there is a consensus among reviewers that the paper is interesting and worthy of publication.  That said, confidence scores are somewhat low, and reviews are fairly brief with limited discussion.  As a result, and given the niche setting assumed by this work, I am only weakly suggesting acceptance for this work.